# Dynamic change of electrostatic field in TMEM16F permeation pathway shifts its ion selectivity

**Wenlei Ye[1]\*, Tina W Han[1], Mu He[1], Yuh Nung Jan[1,2,3], Lily Yeh Jan[1,2,3]\***

[1]Department of Physiology, University of California, San Francisco, San Francisco, United States; [2]Howard Hughes Medical Institute, San Francisco, United States; [3]Department of Biochemistry and Biophysics, University of California, San Francisco, San Francisco, United States

**Abstract** TMEM16F is activated by elevated intracellular $Ca^{2+}$, and functions as a small-conductance ion channel and as a phospholipid scramblase. In contrast to its paralogs, the TMEM16A/B calcium-activated chloride channels, mouse TMEM16F has been reported as a cation-, anion-, or non-selective ion channel, without a definite conclusion. Starting with the Q559K mutant that shows no current rundown and less outward rectification in excised patch, we found that the channel shifted its ion selectivity in response to the change of intracellular $Ca^{2+}$ concentration, with an increased permeability ratio of $Cl^-$ to $Na^+$ ($P_{Cl^-}/P_{Na^+}$) at a higher $Ca^{2+}$ level. The gradual shift of relative ion permeability did not correlate with the channel activation state. Instead, it was indicative of an alteration of electrostatic field in the permeation pathway. The dynamic change of ion selectivity suggests a charge-screening mechanism for TMEM16F ion conduction, and it provides hints to further studies of TMEM16F physiological functions.
DOI: https://doi.org/10.7554/eLife.45187.001

**\*For correspondence:**
wenlei.ye@ucsf.edu (WY);
Lily.Jan@ucsf.edu (LYJ)

**Competing interests:** The authors declare that no competing interests exist.

## Introduction

Mammalian TMEM16F (Anoctamin-6, ANO6) is a membrane protein with dual functions of phospholipid scrambling and ion conduction, both activated by elevation of intracellular $Ca^{2+}$ (*Whitlock and Hartzell, 2017*; *Falzone et al., 2018*; *Bevers and Williamson, 2016*; *Pedemonte and Galietta, 2014*). When activated, TMEM16F mediates the exposure of phosphatidylserine, a lipid normally restricted to the inner leaflet of cell membrane lipid bilayer, to the cell surface (*Suzuki et al., 2010*). This process, known as lipid scrambling, initiates many physiological processes such as recruitment of tissue factors to the platelet surface for thrombin production to trigger blood coagulation and modulation of immune responses in T lymphocytes (*Suzuki et al., 2010*; *Yang et al., 2012*; *Hu et al., 2016*; *Zaitseva et al., 2017*). TMEM16F is also a $Ca^{2+}$-activated small-conductance ion channel (*Yang et al., 2012*), raising the possibility that it has additional physiological functions that have not been revealed. However, there is discrepancy regarding TMEM16F ion selectivity reported by different labs. Recorded with excised-patch inside-out configuration, TMEM16F channels are quickly activated by micromolar $Ca^{2+}$ and they are more permeable to cations (mainly physiological cations such as $Na^+$, $K^+$ and $Ca^{2+}$) than to $Cl^-$ (*Yang et al., 2012*; *Alvadia et al., 2019*). Surprisingly, many groups have reported that the TMEM16F whole-cell current is activated several minutes after cytoplasmic $Ca^{2+}$ elevation, and it displays less cation-selectivity (*Yu et al., 2015*) or even higher permeability to $Cl^-$ than to $Na^+$ (*Grubb et al., 2013*; *Scudieri et al., 2015*; *Shimizu et al., 2013*; *Tian et al., 2012*). This disagreement hampers our further understanding of the functions of this membrane protein.

TMEM16F belongs to the mammalian TMEM16 membrane protein family with 10 members (*Whitlock and Hartzell, 2017*; *Falzone et al., 2018*; *Pedemonte and Galietta, 2014*). The founding members TMEM16A (ANO1) and TMEM16B (ANO2) represent the only two canonical $Ca^{2+}$-activated $Cl^-$ channels (CaCC) without lipid scrambling functions (*Schroeder et al., 2008*; *Caputo et al., 2008*; *Yang et al., 2008*; *Stephan et al., 2009*), while several other mammalian homologs show a modest selectivity between cations and anions but with lipid scrambling capacities (*Suzuki et al., 2010*; *Suzuki et al., 2013*; *Gyobu et al., 2017*; *Le et al., 2019a*; *Watanabe et al., 2018*), a property closer to that of their fungal homologues (*Malvezzi et al., 2013*; *Brunner et al., 2014*; *Lee et al., 2016*). Mammalian TMEM16 proteins are dimeric, as revealed by structural analyses of TMEM16A, TMEM16F and TMEM16K via electron cryo-microscopy (cryo-EM) or crystallography (*Alvadia et al., 2019*; *Paulino et al., 2017a*; *Paulino et al., 2017b*; *Bushell, 2018*; *Dang et al., 2017*; *Feng et al., 2019*), and each subunit contains a permeation pathway that works independently (*Paulino et al., 2017b*; *Lim et al., 2016*; *Jeng et al., 2016*). For TMEM16A, each subunit contains 10 transmembrane helices (TM1 ~TM10), and the stabilization of TM6 by the binding of $Ca^{2+}$ ions to the binding-pocket formed by acidic residues on TM6, TM7 and TM8 within the transmembrane domain opens the permeation pathway, composed of TM3-TM8 (*Paulino et al., 2017a*; *Peters et al., 2018*). Recent studies suggested that, in addition to the pore-lining residues on TM3-TM8 along the TMEM16A permeation pathway for anion conduction, the positive electrostatic field introduced by the $Ca^{2+}$ ions bound to the binding-pocket also contributes to anion accessibility to the pore (*Lam and Dutzler, 2018*). This finding raised the possibility that TMEM16 proteins might adopt a strategy of utilizing the electrostatic field to control ion accessibility to the pore, a process that allows these proteins to be modulated by their surroundings such as $Ca^{2+}$ level. We hypothesize that TMEM16F shifts its ion selectivity in response to elevation of intracellular $Ca^{2+}$ concentrations. However, because TMEM16F current in inside-out excised patch exhibits rapid desensitization and rundown in high $Ca^{2+}$ (*Ye et al., 2018*), it is challenging to test for TMEM16F ion permeability ratio under a wide range of $Ca^{2+}$ concentrations.

In TMEM16F, glutamine 559 (Q559) faces the ionic permeation pathway, and lysine substitution of this pore-lining residue (Q559K) reduces the ratio of $Na^+$ permeability to $Cl^-$ permeability ($P_{Na+}/P_{Cl-}$) (*Yang et al., 2012*; *Alvadia et al., 2019*; *Feng et al., 2019*). Previous studies also show that the current of Q559K persists with prolonged exposure of the excised patch to high intracellular $Ca^{2+}$, and that it displays reduced outward rectification (*Alvadia et al., 2019*; *Nguyen et al., 2019*). In this study, we found that this mutant has different ratio of $Na^+$ permeability to $Cl^-$ permeability ($P_{Na+}/P_{Cl-}$) in different $Ca^{2+}$ concentrations. The shift of permeability ratio does not correlate with alterations of channel open states, but instead is regulated by the change of electrostatic field along the permeation pathway, on which divalent cations such as $Ca^{2+}$ and $Zn^{2+}$ have more significant impact than monovalent ions. Depolarization, which facilitates intracellular cation entry into the membrane electric field, promotes the shift of the relative permeability toward a preference for $Cl^-$ over cations. Such an electrostatic modulation could reflect a general feature of the mechanism of ionic transportation by TMEM16 proteins, and it suggests that TMEM16F harbors an inherent machinery that allows it to dynamically modulate preference between cations and anions in response to its local environment.

## Results

### TMEM16F Q559K increases permeability to $Cl^-$ as intracellular $Ca^{2+}$ increases

Recording from inside-out membrane patch held at +80 mV revealed that wild-type mouse TMEM16F current was activated by intracellular $Ca^{2+}$ in a dose-dependent manner. The current started to decrease when $Ca^{2+}$ was higher than ~30 μM, as a result of both desensitization and rundown (*Figure 1A,C,D*) (*Ye et al., 2018*). Here, desensitization refers to a decreased sensitivity to $Ca^{2+}$, caused by degradation of $PIP_2$ via membrane-tethered phospholipase activated by high intracellular $Ca^{2+}$. Desensitization is also reflected by the requirement of stronger depolarization for channel activation, as a result of the synergy between depolarization and $Ca^{2+}$ level (to be shown later). Rundown refers to a reduction of current magnitude (induced by 1 mM $Ca^{2+}$ in this case). To measure the voltage dependence of activation, we recorded with a voltage-family protocol from

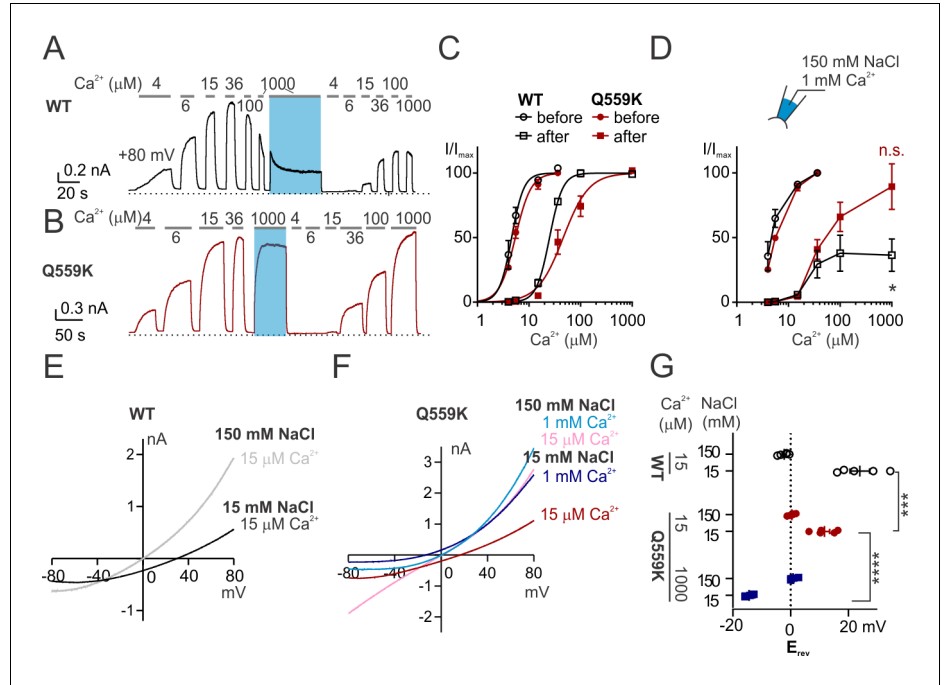

**Figure 1.** TMEM16F Q559K shifts its reversal potential in response to change of intracellular $Ca^{2+}$ concentration. (A, B) Representative recordings of TMEM16F wild type (WT) and Q559K in different $Ca^{2+}$ concentrations. Traces were recorded from transfected HEK293 cells and the inside-out patches were held at +80 mV. The shades illustrate 1 min treatment with 1 mM $Ca^{2+}$ that catalyzes $PIP_2$ degradation by membrane-tethered phospholipase. (C) Dose-response curves for $Ca^{2+}$-activation of WT and Q559K before and after 1 mM $Ca^{2+}$ treatment, respectively. Currents before and after 1 mM $Ca^{2+}$ were separately fitted to the Hill equation and normalized to their respective maximal amplitudes. (D) Change of current magnitudes of WT and Q559K. The currents were normalized to the maximal magnitudes before 1 mM $Ca^{2+}$ for each cell. *$p<0.05$ (one sample $t$ test against hypothetical value '1'). (E, F) Representative I-V relationships of WT and Q559K recorded in indicated conditions. The traces were recorded with a hyperpolarizing ramp from +80 mV to −80 mV (−1 V/s) following holding at +80 mV. (G) Scatter plot of reversal potentials ($E_{rev}$) obtained from traces as in E and F without background correction. $p$-Values were determined with Sidak's multiple comparisons following two-way ANOVA.

DOI: https://doi.org/10.7554/eLife.45187.002

The following figure supplements are available for figure 1:

**Figure supplement 1.** Voltage-dependence of TMEM16F WT and Q559K steady-state activation.
DOI: https://doi.org/10.7554/eLife.45187.003

**Figure supplement 2.** Recordings in indicated conditions with a hyperpolarizing ramp protocol.
DOI: https://doi.org/10.7554/eLife.45187.004

**Figure supplement 3.** Additional control experiments for *Figure 1*.
DOI: https://doi.org/10.7554/eLife.45187.005

**Figure supplement 4.** Permeability ratio $P_{Na+}/P_{Cl-}$ was altered in WT TMEM16F and Q559W in response to change of $Ca^{2+}$.
DOI: https://doi.org/10.7554/eLife.45187.006

**Figure supplement 5.** $I^-$ permeability of TMEM16F WT and Q559K.
DOI: https://doi.org/10.7554/eLife.45187.007

---

−40 mV to +160 mV with 10 mV increments followed by holding at 100 mV to obtain 'tail-currents', which we used as indicators of steady-state conductance at the range of voltages. Fitting to the sigmoidal conductance-voltage (G-V) relationship yielded $V_{1/2}$ of 76 ± 11 mV in 15 μM $Ca^{2+}$ for wild-type TMEM16F, while in 1 mM $Ca^{2+}$, desensitization strongly reduced the voltage-gating, as was previously reported (*Figure 1—figure supplement 1A,B,C*) (*Ye et al., 2018*). The combination of rundown and desensitization rendered it difficult to record the reversal potential ($E_{rev}$) of wild-type TMEM16F in 1 mM $Ca^{2+}$. In our experiments, we measured the $E_{rev}$ from the current recorded with a hyperpolarizing ramp (from +80 mV to −80 mV, 1 V/s) following holding the excised patch at +80

mV. When we switched the bath (equivalent to intracellular solution) from 150 mM NaCl to 45 mM or 15 mM NaCl (osmolarity balanced with mannitol), the current recorded from *Tmem16f*-transfected cells immediately diminished with the hyperpolarizing voltage and was indistinguishable from the background current endogenous to HEK293 cells (*Figure 1—figure supplement 2E,F*). For TMEM16F current induced by 15 µM $Ca^{2+}$ before desensitization, despite the channel closing with the hyperpolarizing voltage, the remaining current was still large enough, distinguishable from the current recorded in 0 $Ca^{2+}$ (referred to as 'background current', *Figure 1—figure supplement 2B, C*). Consistent with previous reports (*Yang et al., 2012*; *Alvadia et al., 2019*), it was moderately selective for $Na^+$ over $Cl^-$ (*Figure 1E*).

In contrast, the mutant Q559K current recorded at +80 mV showed minimal rundown in 1 mM $Ca^{2+}$, in spite of desensitization to $Ca^{2+}$ activation (*Figure 1B*) (*Alvadia et al., 2019*; *Ye et al., 2018*). Normalized to the respective maximal magnitudes of the currents before and after 1-min-treatment of 1 mM $Ca^{2+}$, $EC_{50}$ of $Ca^{2+}$-activation was shifted from 5.6 ± 0.3 µM to 52 ± 11 µM (*Figure 1C*), while the normalized fully activated current magnitude was 0.89 ± 0.18 (current in 1 mM $Ca^{2+}$ normalized to that before 1 mM $Ca^{2+}$), significantly different from 0.36 ± 0.13 for wild type (*Figure 1D*). We also measured the steady-state conductance-voltage relationship from −40 mV to +160 mV. Desensitization caused a right-shift of the voltage-dependence of Q559K current in 15 µM $Ca^{2+}$ (according to the comparison of 15 µM $Ca^{2+}$-activated current before and after 1 mM $Ca^{2+}$-treatment, *Figure 1—figure supplement 1E,G*), but in 1 mM $Ca^{2+}$ the left-shift overrode the effect of desensitization, with $V_{1/2}$ being 18 ± 10 mV (*Figure 1—figure supplement 1F,H*). The left shift of voltage-dependence in 1 mM $Ca^{2+}$ probably explains the absence of rundown in Q559K current as observed for wild type, since at +80 mV Q559K in 1 mM $Ca^{2+}$ is activated to a greater level than Q559K in 15 µM $Ca^{2+}$ (with endogenous $PIP_2$ in cell membrane) or wild type in 1 mM $Ca^{2+}$ (*Figure 1—figure supplement 1*). Taken together, the Q559K mutation that minimized rundown allowed us to record TMEM16F current around physiological membrane potentials in a wide-range of $Ca^{2+}$ concentration, here particularly, to measure $E_{rev}$ in 1 mM $Ca^{2+}$.

Interestingly, the $E_{rev}$ of TMEM16F Q559K current activated by 1 mM $Ca^{2+}$ exhibited a left-shift to −14 ± 1 mV in 15 mM NaCl, suggesting that it was more permeable to $Cl^-$ than to $Na^+$ (*Figure 1F,G*), in contrast to 12 ± 2 mV as the $E_{rev}$ of it in 15 µM $Ca^{2+}$ (recorded with the same protocol as for wild type above). Q559K mutant channels in 1 mM $Ca^{2+}$ had a lower threshold for voltage activation than in 15 µM $Ca^{2+}$ (*Figure 1—figure supplement 1H*), but the current quickly diminished with the hyperpolarizing voltage as in the case of wild-type TMEM16F (*Figure 1—figure supplement 2K*). The remaining current (confirmed with the 'tail current' recorded at +80 mV immediately following the ramp, *Figure 1—figure supplement 2L*) was nonetheless distinguishable from the background current endogenous to HEK293 cells, revealing that it reversed at a negative membrane potential. The current endogenous to HEK293 cells was partially inhibited in 1 mM $Ca^{2+}$ (*Figure 1—figure supplement 3A,B,C*), thus rendering it difficult to measure for background correction in each recording in 1 mM $Ca^{2+}$. However, with the confirmation that the endogenous current was selective for cation (*Figure 1—figure supplement 3A,B,C*) so that the 'real' TMEM16F-mediated current in 1 mM $Ca^{2+}$ should reverse at an even lower membrane potential if the contribution of background current could be adequately removed (*Figure 1—figure supplement 3F*), we chose to take a 'safer' step by not performing background subtraction if the comparison involved currents activated by 1 mM $Ca^{2+}$. Corrected for the liquid junction potential calculated with Clampex, these results indicated that the $P_{Na+}/P_{Cl-}$ was 0.47 ± 0.03 in 1 mM $Ca^{2+}$, in contrast to 2.1 ± 0.4 as calculated from $E_{rev}$ in 15 µM $Ca^{2+}$.

We asked whether the phenotype of $E_{rev}$ shifting is specific to the introduction of lysine to the position Q559 in TMEM16F. For the data shown above, $E_{rev}$ measurements were made after the solution exchange was complete and the current alteration reached equilibrium, in which case the $E_{rev}$ of wild-type TMEM16F in 1 mM $Ca^{2+}$ was not measurable. Now we switched to 1 mM $Ca^{2+}$ when the patch was incubated in 15 mM NaCl and immediately recorded $E_{rev}$. With this protocol, we found a left-shift of $E_{rev}$ accompanied with rapid current inactivation, until the current was too small to be distinguishable from endogenous current (*Figure 1—figure supplement 4A,B,C*). Thus, wild-type TMEM16F dynamically underwent the transition of an increased preference for $Cl^-$ permeation with the elevation of intracellular $Ca^{2+}$ concentration. It has been recently reported that mutation of Q559 to aromatic amino acids (such as Q559W) in TMEM16F also prevents current rundown (*Nguyen et al., 2019*). We measured the $E_{rev}$ of Q559W current activated by 15 µM $Ca^{2+}$ and 1 mM

$Ca^{2+}$ in 15 mM NaCl with the protocol as for Q559K. Although in each condition the $E_{rev}$ of Q559W was more positive than that of Q559K, the $E_{rev}$ shift persisted, regardless of background subtraction (with background currents recorded in $Ca^{2+}$-free solutions with the same NaCl concentration) (*Figure 1—figure supplement 4D,E,F*). The above results showed that TMEM16F increased the relative permeability to $Cl^-$ (versus $Na^+$) when $Ca^{2+}$ concentration was raised from 15 μM to 1 mM, in both wild type and mutants.

TMEM16F has been reported to be permeable to a variety of cations. With the respective published experimental settings, the permeabilities to many physiological cations ($Na^+$, $K^+$, $Ca^{2+}$) were higher than that to $Cl^-$, while the permeability to some other cations may be lower (*Yang et al., 2012*; *Yu et al., 2015*). Among anions, both wild-type TMEM16F and mutant channels are more permeable to $I^-$ than $Cl^-$ (*Figure 1—figure supplement 5*) (*Nguyen et al., 2019*), similar to the preference of CaCC for larger anions (*Schroeder et al., 2008*; *Caputo et al., 2008*; *Yang et al., 2008*). In this study, we will mainly focus on the comparison between $Na^+$ permeability and $Cl^-$ permeability unless otherwise stated. Given the dynamic $P_{Na+}/P_{Cl-}$ ratio, we could not employ the calculation methods normally used for bi-ionic conditions to analyze data obtained with solutions involving a third ion species. Additionally, for each experiment in the following study, we performed paired-comparison within the same patches to avoid ambiguities arising from large variations across different recordings.

## Q559K channel permeability ratio $P_{Na+}/P_{Cl-}$ varies with intracellular $Ca^{2+}$ concentration

We wondered whether the two types of ion selectivity of the Q559K mutant channel correspond to multiple open states. For example, the channel may be in an intermediate open state in 15 μM $Ca^{2+}$ but a fully open state in 1 mM $Ca^{2+}$. To test this possibility, we first tested the ion permeability ratio of current activated by 6 μM $Ca^{2+}$, a condition where both WT and Q559K channels were activated to yield about half of the maximal current magnitudes (*Figure 1A,B,C*). To obtain accurate measurements, we subtracted the current recorded in $Ca^{2+}$-free solution (with 15 mM NaCl) from those recorded in 6 and 15 μM $Ca^{2+}$ for each excised patch. We found that for each recording from either WT or Q559K, the current evoked by 6 μM $Ca^{2+}$ reversed at a more positive voltage than that by 15 μM $Ca^{2+}$ (*Figure 2A,B,C*). To dissociate the effect of $Ca^{2+}$ concentration from that attributable to different open states of the channel, we made use of the $Ca^{2+}$-binding-site mutant, E667Q, that exhibits an elevation of $EC_{50}$ of $Ca^{2+}$-activation (*Yang et al., 2012*; *Alvadia et al., 2019*). The $EC_{50}$ of the double-mutant, Q559K_E667Q, was $0.88 \pm 0.06$ mM, suggesting that the double mutant should not be in the fully open state in 1 mM $Ca^{2+}$ (*Figure 2D,E*). If the increase of anion permeability only occurs when channel is fully open, the current through Q559K_E667Q channels should reverse at a positive potential in 1 mM $Ca^{2+}$. Strikingly, the current in 1 mM $Ca^{2+}$ reversed at $-6.9 \pm 2.0$ mV, corresponding to a channel with higher permeability to $Cl^-$ than to $Na^+$ (*Figure 2F*). This indicates either that TMEM16F ion permeability preference is not correlated with its open state, or that the E667Q mutation circumvents the intermediate open state (if any) and causes the mutant channel to directly enter a fully-open conformation.

To dissociate the correlation between $Ca^{2+}$ concentration and activation state, we utilized the desensitization phenotype of TMEM16F $Ca^{2+}$-gating via exposure to high $Ca^{2+}$. We previously reported that TMEM16F in 36 μM $Ca^{2+}$ undergoes a slow desensitization with time constant of ~60 s possibly due to the slow degradation of $PIP_2$. TMEM16F Q559K activation by 36 μM $Ca^{2+}$ reached full or less-than-half open state, respectively, before or after desensitization (*Figure 1B,C*). We maintained the inside-out patches from Q559K-transfected cells in 36 μM $Ca^{2+}$ (15 mM NaCl) for 90–100 s and measured $E_{rev}$ once every 10 s (*Figure 3*). The $E_{rev}$ was left-shifted when the $Ca^{2+}$ concentration was raised from 15 μM to 36 μM. During the slow desensitization in 36 μM $Ca^{2+}$, $E_{rev}$ (corrected with subtraction of background current recorded in 0 $Ca^{2+}$ afterwards) remained unchanged despite the constant decrease of current magnitude and the increase of outward rectification. We therefore concluded that the reversal potential of Q559K shifted only when $Ca^{2+}$ concentration was changed, independently of current magnitude, open state, or rectification level. Also, although it was not clear whether the permeation pathway could be partially composed of the headgroups of lipids being scrambled, we inferred that at least $PIP_2$, which was slowly depleted during incubation in 36 μM $Ca^{2+}$, did not contribute to the shift of ion permeability ratio.

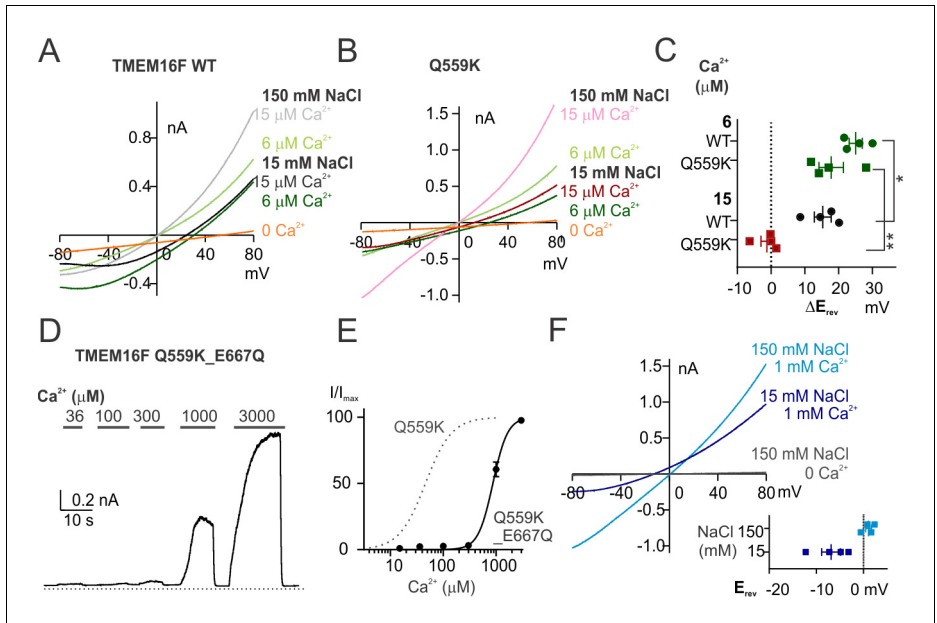

**Figure 2.** TMEM16F Q559K_E667Q channel is more permeable to Cl$^-$ than to Na$^+$ in 1 mM Ca$^{2+}$ despite being half activated. (**A, B**) Representative I-V relationships of WT and Q559K currents recorded in indicated conditions. The recording protocol was the same as in *Figure 1E,F*. Ca$^{2+}$-free 15 mM NaCl solution was applied at the end for background subtraction. (**C**) Scatter plot of the changes of reversal potentials ($\Delta E_{rev}$) when solution was switched to 15 mM NaCl, obtained from traces as in *A* and *B*. *p*-Values were determined with Fisher's LSD test after two-way ANOVA. (**D**) Representative recordings of TMEM16F Q559K_E667Q in different Ca$^{2+}$ concentrations. The recording protocol was the same as in *Figure 1A*. (**E**) Dose-response curve for Ca$^{2+}$-activation of Q559K_E667Q. The gray dotted line represents the curve for Q559K after 1 mM Ca$^{2+}$, replotted from *Figure 1C*. (**F**) Representative I-V relationships of Q559K_E667Q recorded in indicated conditions. The recording protocol was the same as in *Figure 1E,F*. The insert shows the scatter plot of reversal potentials ($E_{rev}$) obtained from traces as in *F*.

DOI: https://doi.org/10.7554/eLife.45187.008

We investigated whether the shift of relative ion permeability involves the conformational change of TM6, a critical step in channel activation. In TMEM16A, the glycine in TM6 (G640 or G644 depending on isoform) works as a hinge to allow the rearrangement of the helical segments during channel activation to generate the ionic permeation pathway (*Paulino et al., 2017a*; *Peters et al., 2018*). Alanine substitution of this glycine stabilizes TM6 at the open conformation and increases TMEM16A Ca$^{2+}$ sensitivity (*Paulino et al., 2017a*; *Peters et al., 2018*). This glycine hinge in TMEM16A TM6 corresponds to G615 in TMEM16F according to sequence alignment (*Figure 4—figure supplement 1A*). Cryo-EM studies reveal that TM6 is bent at this glycine hinge in Ca$^{2+}$-free but not Ca$^{2+}$-bound TMEM16F (*Feng et al., 2019*). Previous studies have shown that its alanine substitution (G615A) increases Ca$^{2+}$ sensitivity (*Alvadia et al., 2019*; *Han et al., 2019*), and we confirmed that G615A left-shifted the voltage dependence of channels recorded in 15 µM Ca$^{2+}$ (*Figure 4A,D*). In contrast, mutation of the adjacent glycine, G614A, did not have similar or further effects (*Figure 4—figure supplement 1E,F,G*). We also confirmed the effect of G615A mutation by comparing its voltage-dependent activation with another two TM6 mutants, I612A and N620A, which were chosen because the alanine substitutions of their corresponding amino acids in TMEM16A (I637A or I641A, Q645A or Q649A, depending on isoform) represent the stabilization of the two steps of TM6 conformational rearrangement during TMEM16A activation, respectively (*Peters et al., 2018*). TMEM16F I612A, mutation of the isoleucine in TM6 upper segment, caused shifts in Ca$^{2+}$- and voltage-gating by an extent comparable with those of G615A (*Figure 4—figure supplement 1D,G*) (*Han et al., 2019*), indicative of a relationship similar to that between TMEM16A G644A and I641A (amino acid labels using TMEM16A isoform as in *Lam and Dutzler, 2018*; equivalent to G640A and I637A for the isoform as in *Peters et al., 2018* ), indicating that the two mutants might be stabilized

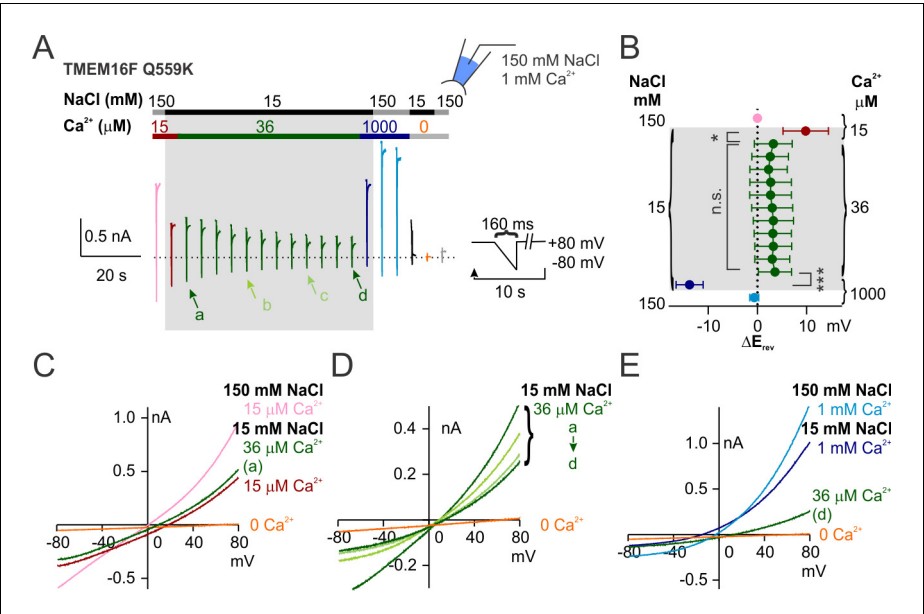

**Figure 3.** Q559K $E_{rev}$ only shifts with the change of $Ca^{2+}$ concentration. (**A**) Representative recording of TMEM16F Q559K held at +80 mV with a hyperpolarizing ramp (−1 V/s) once every 10 s in indicated conditions. (**B**) Summary of the averaged $E_{rev}$ with background subtraction (subtraction of the current in $Ca^{2+}$-free 15 mM NaCl solution, orange trace in *A*, except for the current in 1 mM $Ca^{2+}$) in indicated conditions. For all the traces recorded in 15 mM NaCl, $E_{rev}$ only shifts when $Ca^{2+}$ concentration is changed. (**C, D, E**) Representative I-V relationships of Q559K at arrowed points in *A*, showing that $E_{rev}$ shifts when $Ca^{2+}$ is increased from 15 μM to 36 μM, persists in 36 μM $Ca^{2+}$ despite the constant rundown, and shifts when $Ca^{2+}$ is increased from 36 μM to 1 mM.

DOI: https://doi.org/10.7554/eLife.45187.009

in the same state. In the lower segment, TMEM16A Q649 (Q645 in *Peters et al., 2018*), whose alanine substitution facilitates channel opening (*Peters et al., 2018*; *Lam and Dutzler, 2018*), corresponds to a gap of the alignment in TMEM16F (*Figure 4—figure supplement 1A*); alanine substitution of the asparagine '5-amino-acid-away' in TMEM16F TM6, N620, did not facilitate $Ca^{2+}$ (*Han et al., 2019*) or voltage gating (*Figure 4—figure supplement 1B,C*). Taken together, mutation of G615 to alanine (G615A) might sufficiently stabilize TM6, which in wild-type TMEM16F undergoes rearrangement to generate the ionic permeation pathway.

Notably, the steady-state voltage dependence of TMEM16F G615A channel in 15 μM $Ca^{2+}$ was comparable to that of Q559K in 1 mM $Ca^{2+}$ (*Figure 4A,D*). We then generated the double-mutant Q559K_G615A, which in 1 mM $Ca^{2+}$ showed further left-shift of G-V relationship (*Figure 4B,C,D*). Recorded with the fast hyperpolarizing ramp, the current was almost linear, consistent with the notion that removal of the glycine hinge stabilizes the open conformation (*Figure 4—figure supplement 2*). The change of the permeability ratio $P_{Na+}/P_{Cl-}$ with rising $Ca^{2+}$ in this double mutant was similar to that of Q559K (and wild-type) TMEM16F (*Figure 4E,F*). This double mutant revealed that in the presence of Q559K mutation, TMEM16F channel in 15 μM $Ca^{2+}$ was modestly more permeable to $Na^+$, but it became more permeable to $Cl^-$ in 1 mM $Ca^{2+}$ regardless of TM6 stabilization. Taken together, these results suggest that TMEM16F ion selectivity more likely depends on intracellular $Ca^{2+}$ concentration directly rather than its open state(s), although it remains possible that TMEM16F might undergo miniscule conformational changes at different $Ca^{2+}$ concentrations that contribute to the transition of relative permeability.

## The shift of permeability ratio correlates with electrostatic change in permeation pathway

Intrigued by the possibility that $Ca^{2+}$ alters anion accessibility to the TMEM16A pore through electrostatic effect (*Lam and Dutzler, 2018*), we hypothesize that the shift of permeability ratio of TMEM16F Q559K in response to elevation of intracellular $Ca^{2+}$ is due to a change of electrostatic

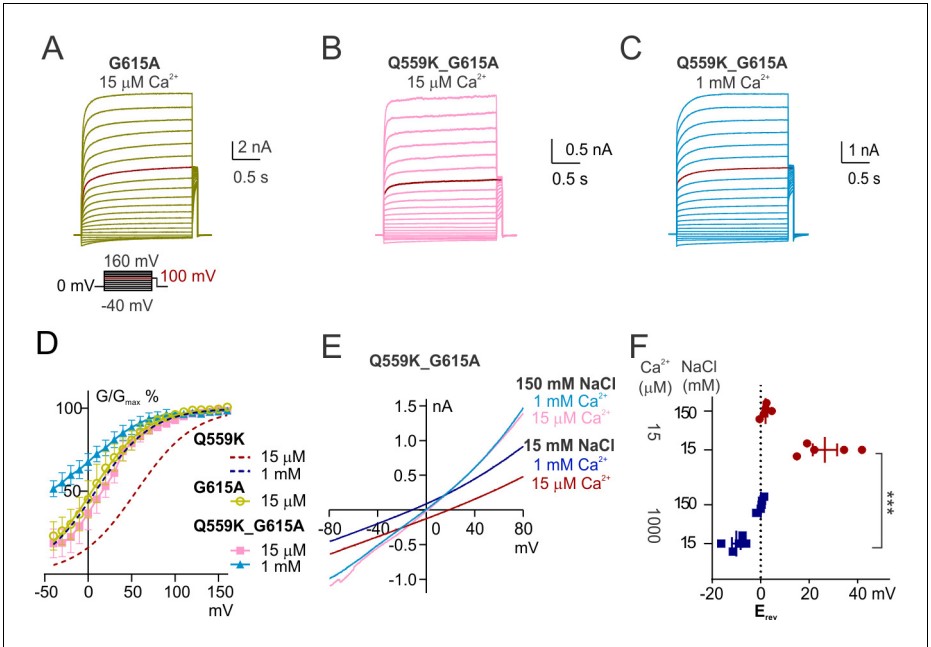

**Figure 4.** The change of ion permeability ratio is preserved despite TM6 conformational stabilization. (**A, B, C**) Representative traces of G615A and Q559K_G615A recorded with a voltage family protocol as in *Figure 1—figure supplement 1*. The currents recorded at +100 mV are highlighted for comparison. (**D**) Averaged G-V relationships of G615A and Q559K_G615A currents. The method for data analysis was the same as that for WT in 15 μM $Ca^{2+}$. The two traces for Q559K were replotted from *Figure 1—figure supplement 1H*. (**D**) Representative I-V relationships of Q559K_G615A recorded in indicated conditions. (**F**) Scatter plot of reversal potentials ($E_{rev}$) obtained from traces as in *E* without background correction. *p*-Values were determined with Sidak's multiple comparisons following two-way ANOVA.

DOI: https://doi.org/10.7554/eLife.45187.010

The following figure supplements are available for figure 4:

**Figure supplement 1.** TM6 functions in TMEM16F gating.
DOI: https://doi.org/10.7554/eLife.45187.011

**Figure supplement 2.** Recordings in indicated conditions with a hyperpolarizing ramp protocol.
DOI: https://doi.org/10.7554/eLife.45187.012

field along its permeation pathway. In this scenario, the effect ought to be elicited not only by $Ca^{2+}$ but also by other divalent or trivalent cations (*Lam and Dutzler, 2018*). We chose to test $Zn^{2+}$ and $Gd^{3+}$, divalent and trivalent cations that are smaller in ionic radius than $Ca^{2+}$ but capable of activating TMEM16F. Both $Zn^{2+}$- and $Gd^{3+}$-activations of TMEM16F were coupled with rapid inactivation even at low concentrations (*Figure 5—figure supplement 1A,D*), precluding the possibility to accurately plot the dose-response curves. The exposure to divalent-free solution (with EGTA) for ~30 s to 1 min following channel inactivation by $Zn^{2+}$ allowed the current to fully recover in magnitude (*Figure 5—figure supplement 1B,C*), suggesting that the rapid current rundown was not attributed to desensitization triggered by $PIP_2$ degradation, although it is unclear whether $Zn^{2+}$ can shield the negative charges in $PIP_2$ headgroups to elicit a reversible rundown.

We found that the wild-type TMEM16F channels activated by 10 μM $Zn^{2+}$ were more permeable to $Na^+$ than to $Cl^-$, as evident from the right-shift in $E_{rev}$ as intracellular NaCl dropped to 15 mM (*Figure 5A*). The $E_{rev}$ in 1 mM $Zn^{2+}$ could not be determined owing to the rapid current inactivation, so we again used the Q559K mutant to this end. Notably, TMEM16F Q559K channels exhibited an increased relative permeability to $Cl^-$ (versus to $Na^+$) with elevation of $Zn^{2+}$ concentration from 10 μM to 1 mM (*Figure 5B,C*). In contrast to the drastic inactivation in 1 mM $Zn^{2+}$, wild-type TMEM16F current activated by $Gd^{3+}$, even with rundown, was still distinguishable from background current at both low and high concentrations. The remaining current underwent the $E_{rev}$ shift when $Gd^{3+}$ concentration was switched from 10 μM to 1 mM (*Figure 5D,E,F*). Note that wild-type TMEM16F

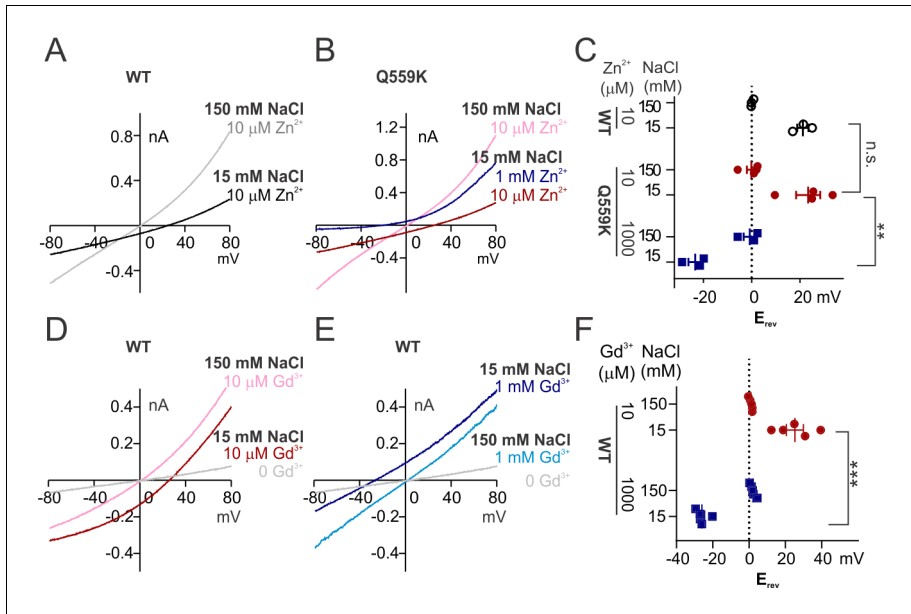

**Figure 5.** The change of ion permeability ratio is preserved when current is activated by $Zn^{2+}$ or $Gd^{3+}$. (A, B) Representative I-V relationships of WT and Q559K recorded in indicated $Zn^{2+}$-containing solutions. The currents were recorded with the same protocol as in *Figure 1E,F*. (C) Scatter plot of reversal potentials ($E_{rev}$) obtained from traces as in *A* and *B*. *p*-Values were determined with Sidak's multiple comparisons following two-way ANOVA. (D, E) Representative I-V relationships of WT TMEM16F recorded in indicated $Gd^{3+}$-containing solutions. Note that the current in 1 mM $Gd^{3+}$ 15 mM NaCl is still distinguishable from background current. (F) Scatter plot of reversal potentials ($E_{rev}$) obtained from traces as in *E* and *F*. *p*-Values were determined with Sidak's multiple comparisons following two-way ANOVA.

DOI: https://doi.org/10.7554/eLife.45187.013

The following figure supplement is available for figure 5:

**Figure supplement 1.** Supplementary recordings for $Zn^{2+}$ and $Gd^{3+}$ activation of TMEM16F.

DOI: https://doi.org/10.7554/eLife.45187.014

current activated by $Gd^{3+}$ showed weaker outward rectification (*Figure 5E*, compared with *Figure 1—figure supplement 3D*). However, the steady-state conductance-voltage relationship (*Figure 5—figure supplement 1E,F*) suggested that, instead of being activated independently of depolarization, the change of rectification was more likely due to a delayed channel closing during hyperpolarization.

Although monovalent ions generate weak electrostatic fields compared with divalent and trivalent ions, given their abundance they should also be able to modulate the electrostatic field along TMEM16F permeation pathway and alter the relative ion permeability. TMEM16F is permeable to most generally-used cations including N-methyl-D-glucamine (NMDG) and tetraethylammonium (TEA) (*Yang et al., 2012*; *Yu et al., 2015*), so introducing any other ion species will cause difficulties in distinguishing whether the shift of $E_{rev}$ is due to the change of $P_{Na+}/P_{Cl-}$ or to the permeation of the new ion species. Thus, we first calculated the respective permeability ratios from the measured $E_{rev}$ with different intracellular NaCl concentrations (*Figure 1—figure supplement 2*). Compared with 15 mM NaCl, the stronger ionic strength of 45 mM intracellular NaCl enhances the screening effect for anions, and thus is predicted to increase the relative permeability to $Cl^-$. Indeed, the calculated $P_{Na+}/P_{Cl-}$ values measured in 15 µM $Ca^{2+}$ were 4.8 ± 0.6 and 2.3 ± 0.2 for wild-type TMEM16F in 15 mM and 45 mM NaCl, respectively; and the calculated $P_{Na+}/P_{Cl-}$ values measured in 15 µM $Ca^{2+}$ were 2.1 ± 0.4 and 1.4 ± 0.1 for Q559K in 15 mM and 45 mM NaCl, respectively (*Figure 6*), consistent with previous findings that Q559K reduces $P_{Na+}/P_{Cl-}$ (*Yang et al., 2012*; *Alvadia et al., 2019*). The $P_{Na+}/P_{Cl-}$ for Q559K in 1 mM $Ca^{2+}$ showed no significant difference between 15 mM NaCl and 45 mM NaCl (*Figure 6*), suggesting that millimolar $Ca^{2+}$ outweighs $Na^+$ in controlling the

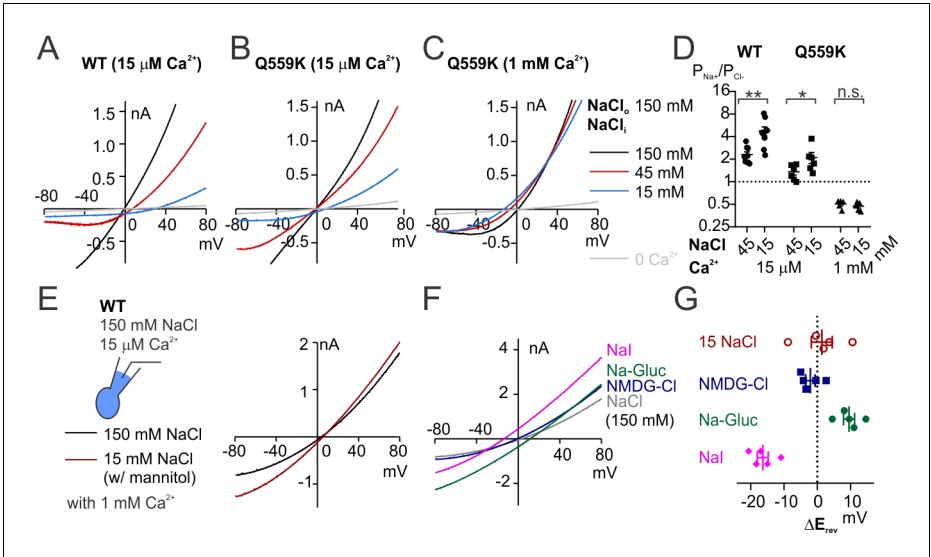

**Figure 6.** Ion permeability ratio is altered with change of intracellular NaCl concentration. (A, B, C) Representative I-V relationships of WT and Q559K recorded with inside-out configuration in indicated conditions. The currents were reanalyzed from the recordings as in *Figure 1—figure supplement 2*. Notice NOT to directly compare the shift of $E_{rev}$ because the intracellular NaCl concentrations are varying. (D) Scatter plot showing the permeability ratio ($P_{Na+}/P_{Cl-}$) calculated from the shift of reversal potentials ($\Delta E_{rev}$) obtained from traces as in *A, B* and *C. p*-Value for WT was determined with Wilcoxon test. *P* values for Q559K were determined with Sidak's multiple comparisons following two-way ANOVA. (E, F) Representative I-V relationships of wild-type TMEM16F currents recorded with whole-cell configuration in indicated bath solutions. NMDG: N-methyl-D-glucamine; Gluc: gluconate. (G) Scatter plot of the changes of reversal potentials ($\Delta E_{rev}$) obtained from recordings as in *E* and *F*. Due to the potentially varying $P_{Na+}/P_{Cl-}$, we did NOT perform statistics or use them to calculate ion permeability ratios.

DOI: https://doi.org/10.7554/eLife.45187.015

The following figure supplements are available for figure 6:

**Figure supplement 1.** Recordings involving 15 mM NaCl bath balanced with NMDG-MES or $NMDG_2-SO_4$.
DOI: https://doi.org/10.7554/eLife.45187.016

**Figure supplement 2.** Recordings with 15 mM NaCl in the pipette solution.
DOI: https://doi.org/10.7554/eLife.45187.017

electrostatic effect. The shift of $E_{rev}$ by changing the intracellular NaCl is also consistent with our proposed scenario that the shift of ion permeability ratio is not due to TM6 conformational change associated with different open states, but instead, due to the alteration of electrostatic field along the permeation pathway.

The minor shift of relative ion permeability by the change of intracellular NaCl concentration shown above might account for the discrepancy regarding TMEM16F ion selectivity recorded with various methods used by different labs. In the experiments above, the osmolarities of low NaCl solutions were all balanced with mannitol to avoid the interference by other ion species. Now, we tested for 15 mM NaCl with equal ionic strength as extracellular solution, which was balanced either with 135 mM NMDG-MES (MES: methanesulfonic acid) or 68 mM $NMDG_2-SO_4$. We observed that the wild-type TMEM16F activated by 15 μM $Ca^{2+}$ in both conditions reversed at ~0 mV (2 ± 1 mV in 15 mM NaCl with NMDG-MES, 3 ± 2 mV in that with $NMDG_2-SO_4$; *Figure 6—figure supplement 1A, C*). Compared with $E_{rev}$ recorded in 15 mM NaCl with mannitol, the reduction of $E_{rev}$ might reflect a combination of NMDG permeation and the reduction of $P_{Na+}/P_{Cl-}$, assuming the permeation of $MES^-$ or $SO_4^{2-}$ was negligible. In neither condition could we determine the $E_{rev}$ for currents activated by 1 mM $Ca^{2+}$. Using the mutant Q559K, we confirmed that in both conditions, the shift of $E_{rev}$ with rising $Ca^{2+}$ persisted (*Figure 6—figure supplement 1B,D,E*), which could be explained by an increase of $P_{Cl-}/P_{Na+}$ or a robustly increased permeation of NMDG, with the latter being less likely.

To confirm that the shift of ion permeability ratio does not depend on recording configuration, we altered the compositions of the pipette solution rather than those of the bath solution. We applied 15 mM NaCl balanced with mannitol, NMDG-MES or $NMDG_2$-$SO_4$ to the pipette solution (equivalent to extracellular solution), while the bath (equivalent to intracellular solution) contained 150 mM NaCl. With the correction of liquid junction potentials (12 mV, 6 mV and 2 mV respectively), the currents reversed at ~0 mV when activated by 15 μM $Ca^{2+}$ in all the conditions ($-1 \pm 1$ mV, $4 \pm 2$ mV and $-1 \pm 2$ mV, respectively), while at $11 \pm 2$ mV, $19 \pm 1$ mV and $24 \pm 1$ mV, respectively, when activated by 1 mM $Ca^{2+}$ (*Figure 6—figure supplement 2*). For the conditions involving NMDG, the $E_{rev}$ shift either indicated an increased $P_{Cl-}/P_{Na+}$ or a robustly increased permeation of NMDG, with the latter being less likely. These results also complemented the previous tests involving changes of intracellular NaCl concentration (*Figure 6A–D*), showing that enhancement of intracellular ionic strength 'unidirectionally' promotes the relative permeability to $Cl^-$. This can be attributed either to the preceding depolarization which drives cations into the pore, or to the possibility that the pore intrinsically adopts cations to modulate the electrostatic field. In addition, there was no significant difference in $E_{rev}$ shifting between solutions balanced with NMDG-MES and that with $NMDG_2$-$SO_4$, suggesting that the divalent anions are relatively inert whether being applied intracellularly or extracellularly, attributable to the inaccessibility of divalent anions to the permeation pathway.

The above results suggest that TMEM16F ion selectivity is modulated by the change of electrostatic field along its permeation pathway, with divalent or trivalent cations having stronger impacts than monovalent ions. This might account for the lower selectivity for $Na^+$ over $Cl^-$ obtained from whole-cell recording, where intracellular solution was usually an isotonic salt solution. In whole-cell recording, TMEM16F current is reported to be elicited a few minutes after the whole-cell configuration is formed, and less selective for cations or in certain conditions even more selective for $Cl^-$. Based on results reported above, we could infer that in these conditions the intracellular ionic strength (mainly maintained by ~150 mM NaCl or KCl) would reduce the channel relative permeability to cations. We performed whole-cell recording with 150 mM NaCl and 15 μM $Ca^{2+}$ in the pipette, the $E_{rev}$ was $1.4 \pm 3.1$ mV in 15 mM NaCl, indicative of $P_{Na+}/P_{Cl-}$ of $1.0 \pm 0.1$ (*Figure 6D,F*). Replacement of extracellular $Na^+$ with NMDG did not significantly shift $E_{rev}$, indicative of an increased permeability to $Cl^-$ and/or a modest permeability to NMDG. Replacement of extracellular $Cl^-$ with gluconate (Gluc) or $I^-$ shifted $E_{rev}$ to positive or negative values respectively, suggesting the presence of $Cl^-$ conduction (*Figure 6E,F*) as observed in previous studies. However, we could not calculate the permeability ratios due to the potentially varying $P_{Na+}/P_{Cl-}$ in these conditions. Notably, it is also reported that *Tmem16f*-transfected cells undergo an 'unconventional exocytosis' which expands the membrane surface area dramatically during recording (*Bricogne et al., 2019*), leading to challenges in performing membrane capacitance compensation. These technical issues rendered it difficult to perform accurate measurements with whole-cell recording.

Taken together, the elevation of intracellular cation level underlies the increased permeability to $Cl^-$, with divalent cations ($Ca^{2+}$ and $Zn^{2+}$) and trivalent cations ($Gd^{3+}$) having stronger effects than monovalent ions ($Na^+$ and NMDG). This suggests that TMEM16F employs a charge-screening mechanism to dynamically alter the preference for permeating ion species: Increase of positive charges, by introduction of basic amino acids or by entry of cations, enhances the channel preference for anions. This mechanism allows TMEM16F to open as a non-selective ion channel at low intracellular $Ca^{2+}$ level and as a $Cl^-$ channel when local $Ca^{2+}$ concentration increases, indicative of physiological functions that have not been investigated thus far.

## Depolarization alters permeability ratio synergistically with intracellular $Ca^{2+}$

Previous studies have shown the synergy between $Ca^{2+}$ and depolarization in the gating of TMEM16A. Depolarization facilitates $Ca^{2+}$ entry into the membrane electric field and thus reducing the $EC_{50}$ for channel activation by $Ca^{2+}$ (*Xiao et al., 2011*; *Peters et al., 2018*). The reduction of $EC_{50}$ of $Ca^{2+}$ activation by depolarization is also observed in TMEM16F (*Yang et al., 2012*; *Ye et al., 2018*), although TMEM16F permeates $Ca^{2+}$ (*Yang et al., 2012*), suggesting $Ca^{2+}$ efflux is not fast enough to deplete $Ca^{2+}$ from the electric field at the intracellular side. We asked whether depolarization also increases TMEM16F permeability to $Cl^-$ by driving $Ca^{2+}$ into the membrane electric field. To this end, we held the excised membrane patch at potentials ranging from +40 to+160 mV with increments of 40 mV (referred to as 'conditioning potentials') followed by a hyperpolarizing ramp

from +80 mV to −80 mV (−2 V/s), to measure the $E_{rev}$ (*Figure 7C*). With this experimental design, the shift of $E_{rev}$ following different conditioning potentials could provide an indication of the changes of relative permeability, under the circumstance that during hyperpolarization TMEM16F can temporarily 'memorize' the ion permeability ratio at the preceding conditioning potential. We measured the $E_{rev}$ of TMEM16F wild-type channels in 15 μM $Ca^{2+}$, Q559K channels in 15 μM and in 1 mM $Ca^{2+}$. In 15 μM $Ca^{2+}$, the more depolarized the conditioning potential, the higher the relative $Cl^-$ permeability (versus $Na^+$ permeability) (*Figure 7A,D*). The traces recorded with the hyperpolarizing ramp following different conditioning potentials did not intersect with each other, confirming the

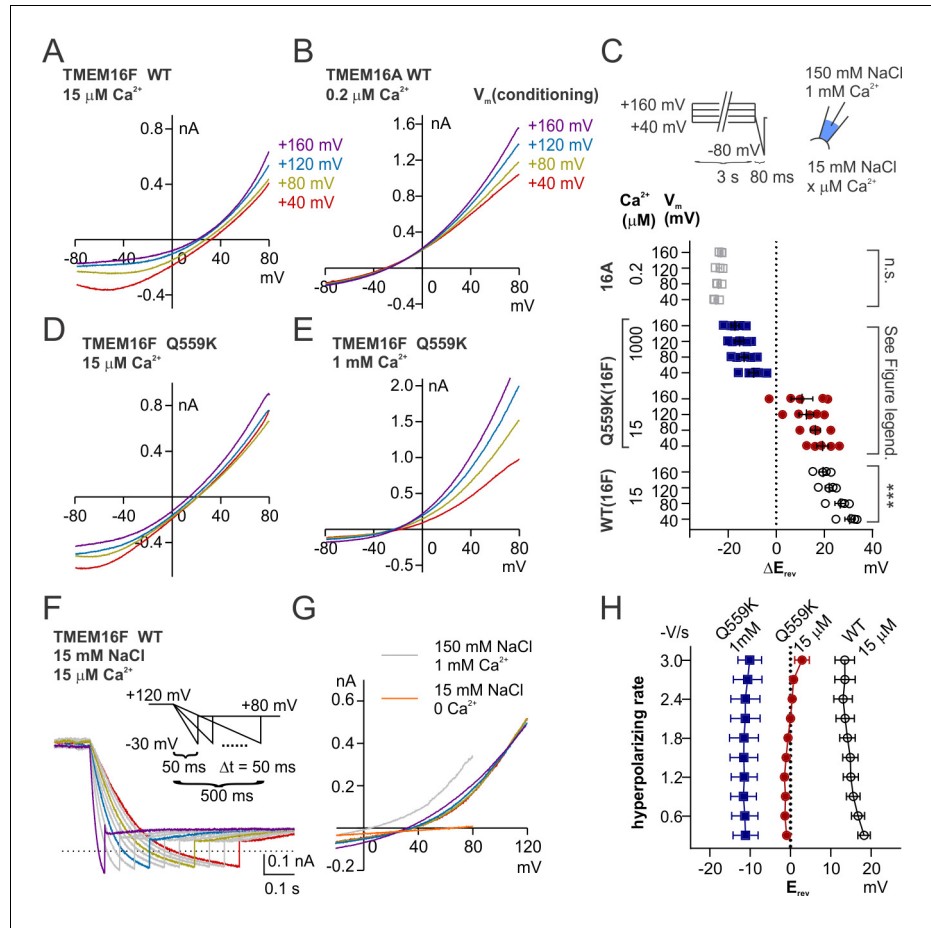

**Figure 7.** Depolarization alters permeability ratio synergistically with $Ca^{2+}$ level. (A, B, D, E) Representative I-V relationships of currents recorded in indicated conditions. The excised patch was held at +40 to+160 mV with increments of 40 mV ('conditioning potentials') followed by a hyperpolarizing ramp from +80 mV to −80 mV (−2 V/ s). (C) Scatter plot showing the changes of reversal potentials ($\Delta E_{rev}$) when solution was switched from 150 mM NaCl to 15 mM NaCl. For TMEM16A and WT TMEM16F, *p*-values were determined with one-way ANOVA. For TMEM16F Q559K, two-way ANOVA shows *p*<0.001 across voltages, *p*<0.0001 across $Ca^{2+}$ concentrations. (F) Representative WT TMEM16F traces recorded in 15 μM $Ca^{2+}$ 15 mM NaCl bath solution with the indicated protocol. The excised patch was held at 120 mV followed by a hyperpolarizing ramp from +120 mV to −30 mV (ramping speed from −0.3 V/s to −3 V/s), and the reversal potentials were corrected with background current recorded at the end. (G) The I-V relationships of the highlighted traces as in *F*, showing the currents reverse at the same point despite the change of rectification. (H). Summary of reversal potentials at different hyperpolarizing speeds obtained from traces as in *G*. Two-way ANOVA suggests that there is no significant difference among various hyperpolarizing speeds (*p*=0.28).

DOI: https://doi.org/10.7554/eLife.45187.018

The following figure supplement is available for figure 7:

**Figure supplement 1.** Representative traces of Q559K recorded with ramps of varoius hyperpolarizing speeds.

DOI: https://doi.org/10.7554/eLife.45187.019

$E_{rev}$ alteration even though we did not perform background subtraction. The shift of $E_{rev}$ was not a potential artifact caused by ion accumulation or depletion as previously investigated for other channels (*Li et al., 2015*; *Yu et al., 2014*), since if it was, a more depolarized conditioning potential would cause more severe $Na^+$ accumulation or $Cl^-$ depletion at the extracellular side (in the pipette solution), leading to a right-shift of $E_{rev}$, which was opposite to our results. Traces of Q559K currents following different conditioning potentials in 1 mM $Ca^{2+}$ overlapped at low membrane voltages (*Figure 7E*), a phenomenon observed for TMEM16A in 0.2 µM $Ca^{2+}$ (*Figure 7B*). The gradual increase of relative $Cl^-$ permeability caused by preceding depolarization can be attributed to intracellular cations, particularly $Ca^{2+}$, to the membrane electric field to enhance the attraction of permeation pathway to anions, as can be inferred from the charge-screening mechanism.

The observation that TMEM16F in 15 µM $Ca^{2+}$ is able to 'memorize' the ion permeability ratio at the preceding conditioning potential, suggests that the electrostatic field along the permeation pathway can be maintained even when there is no movement of net charge (i.e. 0 pA). We then recorded the $E_{rev}$ with hyperpolarizing ramps of variable speeds (0.3 V/s to 3 V/s) following holding the patch at +120 mV, to test whether the channel would 'forget' the ion permeability ratio at the conditioning potential (+120 mV) by observing $E_{rev}$s under hyperpolarization ramps of various rates. We did not see a significant shift of the $E_{rev}$ for both WT and Q559K, although wild-type TMEM16F current became more outwardly rectifying at slower hyperpolarizing ramps (*Figure 7F,G*, *Figure 7— figure supplement 1*). Thus, the permeability ratio ($P_{Cl-}/P_{Na+}$) can be maintained during hyperpolarization, raising the possibility that the $Ca^{2+}$ ion that modulates the pore electrostatic field dwells at the permeation pathway or a proximal site that allows it to still affect permeation pathway instead of leaving the pore with the ionic flow.

We summarized the mechanism of TMEM16F dynamic ion selectivity based on the experiments shown above and published previously (*Figure 8*). TMEM16F ion permeation pathway is opened as a result of protein conformational change, which probably involves the rearrangement of TM6 induced by $Ca^{2+}$-binding to the conserved $Ca^{2+}$-binding pocket. The elevation of intracellular divalent or trivalent cation concentration brings positive charges into the permeation pathway, which gradually increases the relative permeability of anions. Depolarization promotes the pore attraction to anions by facilitating $Ca^{2+}$ entry into the membrane electric field, wherein it may remain until the channel is closed.

## Discussion

Studies regarding TMEM16F physiological functions have mostly focused on its phospholipid scrambling activities, while the understanding of its ion channel functions is confounded by the discrepancy of ion selectivity measurements. In cells with TMEM16F expression, increasing cytoplasmic $Ca^{2+}$ concentration leads to activation of TMEM16F channel that mediates $Ca^{2+}$ influx (*Feng et al., 2019*; *Han et al., 2019*), which further activates phospholipid scrambling. This is consistent with the results that TMEM16F recorded with inside-out excised patch showed higher permeability to cations than to $Cl^-$ (*Yang et al., 2012*; *Alvadia et al., 2019*), notwithstanding the whole-cell recording indicating that delayed-activated TMEM16F is an anion channel (*Grubb et al., 2013*). In light of the Q559K mutant which displays minimal rundown and less outward rectification in 1 mM $Ca^{2+}$, we measured TMEM16F reversal potentials at different $Ca^{2+}$ concentrations, revealing a shift toward $Cl^-$-selectivity at high intracellular $Ca^{2+}$. This shift of ion permeability ratio correlates with the change of electrostatic field in the permeation pathway, which is determined by both intracellular ion concentration and membrane potential; divalent cations regardless of ion species are expected to exert stronger electrostatic effects than monovalent cations. This might reflect a general feature in the coupling of gating and conduction for channels in the TMEM16 family, likely relevant to their physiological functions.

### Gating and ion selectivity of TMEM16F

The ion permeation pathway of TMEM16F is generated probably through the rearrangement of TM6 in response to elevation of intracellular $Ca^{2+}$ concentration, as reflected in its conformation in the presence or absence of $Ca^{2+}$ (*Feng et al., 2019*), a mechanism similar to that of TMEM16A. Further elevation of intracellular $Ca^{2+}$ results in an increase of relative permeability to $Cl^-$ (versus $Na^+$). With membrane potential and ionic strength unchanged, the permeability ratio $P_{Na+}/P_{Cl-}$ is only

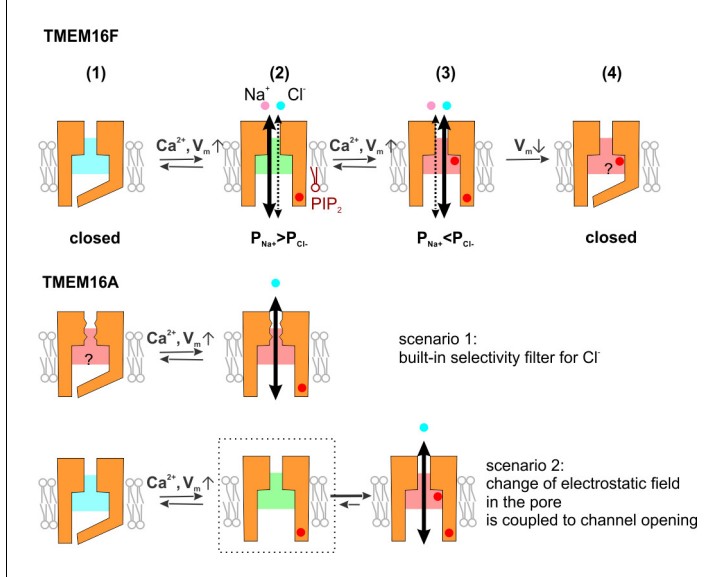

**Figure 8.** Diagram showing the proposed ion selectivity mechanism for TMEM16F and a comparison with TMEM16A. (**Upper**) TMEM16F gating and ion selectivity. The binding of intracellular $Ca^{2+}$ to the binding-pocket triggers the rearrangement of transmembrane helices and opens the ion permeation pathway (from 1 to 2). The elevation of intracellular $Ca^{2+}$ level shifts the electrostatic field along the permeation pathway from a $Na^+$-favoring state (shaded with cold color) to that with increased attraction to $Cl^-$ (shaded with warm color, from 2 to 3). Each red dot represents a $Ca^{2+}$-entry event, while the number of $Ca^{2+}$ ions for every event is not specified. Depolarization drives $Ca^{2+}$ into both the binding pocket and the permeation pathway; $PIP_2$ stabilizes the open conformation but does not affect the ion selectivity. Hyperpolarization triggers the closing of the channel, preceding the retreating of $Ca^{2+}$ from the site that allows it to affect the permeation pathway, so that the channel maintains the electrostatic field and 'memorizes' the permeability ratio (from 3 to 4). (**Lower**) TMEM16A might harbor a selectivity filter restrictedly selective for anions (scenario 1), either with a size-dependent mechanism (illustrated with the zigzags) or a strong electrostatic field (illustrated with red shade). Alternatively, the change of electrostatic field might be tightly coupled to $Ca^{2+}$-gating (scenario 2), so that the pore restrictedly selects for anions once it is opened.

DOI: https://doi.org/10.7554/eLife.45187.020

The following figure supplement is available for figure 8:

**Figure supplement 1.** TMEM16A ion selectivity.
DOI: https://doi.org/10.7554/eLife.45187.021

correlated with $Ca^{2+}$ concentration rather than $Ca^{2+}$-induced conformational change, suggesting that the ion selectivity machinery is not coupled to channel gating. At low $Ca^{2+}$ when channels display modest selectivity between $Na^+$ and $Cl^-$, the ion permeability ratio is also altered by monovalent ions and membrane potential, which indicates that the ion selectivity filter is sensitive to electrostatic disturbance and thus might explain the discrepancy of experimental results from different labs. Depolarization, which drives $Ca^{2+}$ into membrane electric field and facilitates $Ca^{2+}$-gating, synergistically promotes the permeability alteration towards a preference for $Cl^-$. The ion permeability ratio can be memorized during the recording with a hyperpolarizing ramp, suggesting that $Ca^{2+}$ retreating from the site that affects the permeation pathway is preceded by the dissociation of $Ca^{2+}$ from the binding pocket, so that the pore electrostatic field can be maintained until the channel is closed. $PIP_2$ reduces the $EC_{50}$ for $Ca^{2+}$-activation but does not affect ion selectivity, thus facilitating TMEM16F opening when the pore is still more attractive to cations in low intracellular $Ca^{2+}$. Controlling the selectivity between cations and anions by altering the pore potential, referred to as a charge-screening mechanism, is also documented in other ion channels, such as ligand-gated channels (*Keramidas et al., 2004*) and bacterial porins (*Alcaraz et al., 2004*), through charged residues lining the pore and/or through permeating ions within the pore.

It is intriguing to consider the structural basis for $Ca^{2+}$ regulation of the TMEM16F relative ion permeability. Specifically, we wonder whether $Ca^{2+}$ ion(s) in the $Ca^{2+}$-binding pocket, or in the

permeation pathway, may alter the electrostatic field of the pore. The former resembles what has been proposed for TMEM16A (*Lam and Dutzler, 2018*), whose pore is electrostatically modulated by $Ca^{2+}$ bound to the pocket, but it seems unlikely for TMEM16F, because the Hill coefficient of $Ca^{2+}$-activation of TMEM16F is ~2, suggesting that the two $Ca^{2+}$-binding sites should both be occupied when the current displays higher permeability to cations. The latter scenario is that there is room for $Ca^{2+}$-dwelling in the permeation pathway, consistent with the recently-published TMEM16F structure in which the pore harbors a large vestibule (*Alvadia et al., 2019*; *Feng et al., 2019*). Also, its distant homolog, TMEM16K, has multiple $Ca^{2+}$ ions bound to its cytoplasmic domains (*Bushell, 2018*), raising the question whether there could be a reservoir in the TMEM16 cytoplasmic domain to store $Ca^{2+}$. Given the long range electrostatic actions, it will be technically challenging to identify the structural basis for the electrostatic modulation in the pore.

## Comparison among TMEM16 members

TMEM16 represents a family of membrane proteins with diverse functions, including $Ca^{2+}$-activated ion channel activities and $Ca^{2+}$-activated lipid scrambling. Mammalian TMEM16A and TMEM16B are $Ca^{2+}$-activated $Cl^-$ channels (*Schroeder et al., 2008*; *Caputo et al., 2008*; *Yang et al., 2008*; *Stephan et al., 2009*), while other members with phospholipid scrambling functions display modest cation-selective or non-selective ion channel activities, such as TMEM16E (*Whitlock et al., 2018*), TMEM16F (*Yang et al., 2012*; *Alvadia et al., 2019*; *Yu et al., 2015*; *Le et al., 2019a*), *Drosophila* subdued (*Le et al., 2019b*), and two fungal TMEM16 homologs (*Malvezzi et al., 2013*; *Brunner et al., 2014*; *Lee et al., 2016*; *Jiang et al., 2017*). The lipid scrambling activity is also observed in an amoebozoa TMEM16 homolog (*Pelz et al., 2018*). All the TMEM16 proteins, based on reported structures and sequence alignment, are dimers with each monomer containing 10 transmembrane helices with conserved $Ca^{2+}$-binding sites (*Alvadia et al., 2019*; *Brunner et al., 2014*; *Paulino et al., 2017a*; *Bushell, 2018*; *Dang et al., 2017*; *Feng et al., 2019*; *Kalienkova et al., 2019*; *Falzone et al., 2019*). Here, our studies reveal that TMEM16F displays a preference for $Cl^-$ conduction when $Ca^{2+}$ concentration is high, potentially indicative of a unifying mechanism regarding TMEM16 channel functions. In the following paragraphs, we will briefly discuss the similarities and differences among TMEM16 members, and focus mainly on the comparison between TMEM16A and TMEM16F.

In both TMEM16A and TMEM16F, the opening of the permeation pathway involves the conformational stabilization of TM6, while the removal of a glycine hinge reduces the $Ca^{2+}$ level required for activation (*Paulino et al., 2017a*; *Peters et al., 2018*). Alanine substitution of the glycine reduces $EC_{50}$s of $Ca^{2+}$-activation in both TMEM16A and TMEM16F. However, TMEM16A and TMEM16F currents differ in rectification level (*Nguyen et al., 2019*). The current of TMEM16A bound to one $Ca^{2+}$ ion (at low intracellular $Ca^{2+}$ level) is outwardly rectifying, while at high intracellular $Ca^{2+}$ level that allows two $Ca^{2+}$ ions to occupy the binding-pocket, the TMEM16A channel conductance becomes 'Ohmic' (*Peters et al., 2018*). Such a transition can be explained by two mutually compatible mechanisms: the second $Ca^{2+}$ ion triggers the further rearrangement of TM6 (*Peters et al., 2018*) and its positive electrostatic field reduces the energy barrier that intracellular anions need to overcome to access the permeation pathway (*Lam and Dutzler, 2018*). In contrast, TMEM16F current is outwardly rectifying in a wide range of $Ca^{2+}$ concentration. It is important to keep in mind that TMEM16F rectification indices obtained from the traces recorded with ramp protocols (such as in *Figure 1E,F*) can be confounded with the voltage-dependent gating. However, according to the 'instantaneous rectification index' (*Alvadia et al., 2019*), which describes the magnitude ratio of outward current to inward current within the time range only allowing minimal change of gating, the TMEM16F pore is intrinsically outwardly rectifying, and wild type and Q559K channels are likely rectifying to a similar extent (*Alvadia et al., 2019*). Thus, in contrast to the corresponding pore-lining residue K588 in TMEM16A, which electrostatically interacts with the permeating $Cl^-$ and is necessary for the ohmic conductance (*Paulino et al., 2017b*; *Nguyen et al., 2019*), the reduction of outward rectification of TMEM16F Q559K mutation might result from the alteration of voltage gating. Interestingly, the TMEM16F current activated by 1 mM $Gd^{3+}$ is potentially approaching the 'ohmic' state (*Figure 5E*), suggesting that the strong electrostatic field of a trivalent cation bound to the protein could enhance the accessibility of intracellular $Cl^-$ to the pore.

We propose two scenarios to explain why TMEM16F displays a shifting ion selectivity, while TMEM16A selects for $Cl^-$ at all $Ca^{2+}$ concentrations (*Figure 8*). In one scenario, TMEM16A has a

'built-in' selectivity filter that favors anions, so that only anions can eventually permeate even though both anions and cations can access the pore. Such a 'built-in' selectivity filter can either be a size-based screening structure, or a constantly strong positively-charged field to attract anions and repel cations. In the second scenario, TMEM16A couples the change of the electrostatic field to the opening of the pore, in a way that the former event precedes or synchronizes with the latter. We tried to distinguish between the two scenarios by measuring the ion selectivity of TMEM16A activated by low intracellular $Ca^{2+}$. Notably, TMEM16A current activated by 100 nM $Ca^{2+}$ or 300 nM $Ca^{2+}$ is $Cl^-$-selective, with the selectivity measured in low NaCl balanced with NMDG-MES being more selective for $Cl^-$ (compared with $Na^+$) than that in low NaCl balanced with mannitol (*Figure 8—figure supplement 1A,B,C*), a phenomenon consistent with a previous report (*Lim et al., 2016*). To probe TMEM16A ion selectivity in 0 $Ca^{2+}$, we also generated the Q649A mutant of TMEM16A, which is activated by depolarization in $Ca^{2+}$-free solution (*Peters et al., 2018*; *Lam and Dutzler, 2018*), and we used poly-L-lysine (PLL) to strip off membrane-tethered $PIP_2$ to accelerate current rundown (*De Jesús-Pérez et al., 2018*). We noticed that TMEM16A Q649A activated by 30 nM $Ca^{2+}$ was selective for $Cl^-$ before or after PLL application, but the 'rundown' component (current in 0 $Ca^{2+}$) displayed a dispersed distribution of $E_{rev}$ (*Figure 8—figure supplement 1D–I*). Because PLL also triggered the rundown of the background current endogenous to HEK293 cells, which is cation selective (*Figure 8—figure supplement 1J*), we could not draw a definitive conclusion at this point.

## Functional implication

The shift of relative ion permeability with elevation of $Ca^{2+}$ concentration may dynamically regulate membrane potential of the cell. Since TMEM16F can function as a $Ca^{2+}$-activated $Ca^{2+}$-permeable channel, the dynamic increase of $Cl^-$ permeability (versus cation) might provide a brake to the positive feedback to reduce the influx of cations including $Ca^{2+}$. In a typical neuronal cell, such a transition towards $Cl^-$ selectivity might reduce the entry of $Ca^{2+}$ into the cell and drive membrane potential close to $Cl^-$ equilibrium potential, both of which will reduce the neuronal excitability and prevent $Ca^{2+}$-overloading. The 'memory' of the permeability ratio during repolarization allows TMEM16F to modulate the waveform of action potentials in accordance to local cytoplasmic $Ca^{2+}$ level. In non-excitable cells, this function might enable TMEM16F to sense cellular $Ca^{2+}$ level and modulate membrane potential, which affects cellular proliferation and differentiation (*Sundelacruz et al., 2009*). It has also been reported that TMEM16F is localized in primary cilia (*Forschbach et al., 2015*), a special cellular compartment with elevated $Ca^{2+}$ level and greater depolarization (*Delling et al., 2013*), providing a condition for TMEM16F to function dynamically. With the broad expression pattern of TMEM16F, we envision a new perspective to examine the functions of TMEM16F.

## Materials and methods

### Cell culture and molecular biology

Mouse *Tmem16f and Tmem16a* cDNAs (sequences as in NCBI RefSeq NM_175344.4, *Mus musculus Ano6*-splice variant 2, and NM_178642.5, *Mus musculus Ano1*-splice variant 1, respectively) were respectively fused with mCherry and subcloned into pcDNA3, as previously reported (*Ye et al., 2018*). Site-directed mutagenesis was performed using standard molecular techniques with pHusion polymerase (New England Biolabs, Ipswich, MA) and sequences were all verified (Quintara Biosciences, South San Francisco, CA). HEK (Human embryonic kidney)−293 cells (ATCC, RRID: CVCL_0045) were aliquoted and preserved in liquid nitrogen. One vial of cells was thawed and plated in 10-mL cell culture flask (ThermoFisher Scientific, Waltham, MA) once every 3 months. Cell were cultured in Dulbecco's Modified Eagle Medium (DMEM, with 4.5 g/L glucose, L-glutamine and sodium pyruvate, Mediatech, Manassas, VA) containing 10% FBS (Axenia BioLogix, Dixon, CA) and 1% penicillin-streptomycin, at 37°C and with 5% $CO_2$, and they were passaged into a new flask with a 1:10 ratio once every 3 ~ 4 days. They were disposed of if they were not confluent in the flask in 4 days, their morphologies deviated from pictures in instructions (ThermoFisher website), or if the batch had been used for 3 months, whichever coming first, but were not in particular tested for mycoplasma contamination. Cells for electrophysiology recording were plated on 12 mm round coverslip (Warner Instruments, Hamden, CT) during the passaging procedure. Transient transfection was performed with

Lipofectamine 2000 (ThermoFisher Scientific, Waltham, MA) 2 days before recording. The cDNA constructs for wild-type TMEM16F-mCherry, the mutants Q559K-, G615A-, G614A-, I612A-, G614_G615A-, N620A-, and E667Q- mCherry were stably transfected in HEK293 cells as previously reported (*Han et al., 2019*). Briefly, the cDNAs were subcloned into pENTR1A (Addgene plasmid #17398) and transferred to pLenti CMV Hygro DEST (Addgene plasmid #17454) using Gateway cloning (*Campeau et al., 2009*). pENTR1A no ccDB (w48-1) and pLenti CMV Hygro DEST (w117-1) were gifts from Dr. Eric Campeau and Dr. Paul Kaufman (University of Massachusetts Medical School, Worcester, MA, USA). TMEM16F-mCherry pLenti was co-transfected into HEK293FT cells with packaging plasmids pMD.2G and psPAX2, which were gifts from Didier Trono (Addgene plasmids # 12259 and #12260). Lentivirus was harvested from the transfected cells 36–48 hr post-transfection and incubated with HEK293 cells to establish stable cell lines under hygromycin selection.

## Solutions

For all electrophysiology recordings, bath solution contained 145 mM NaCl, 10 mM HEPES, 2 mM $CaCl_2$, 1 mM $MgCl_2$, 10 mM glucose, pH 7.2 with NaOH. For inside-out recordings, pipette solution contained 150 mM NaCl, 10 mM HEPES, 1 mM $CaCl_2$, unless otherwise stated. The membrane patch was excised to form inside-out configuration in $Ca^{2+}$-free solution: 150 mM NaCl, 10 mM HEPES, 2 mM EGTA. For solutions with $Ca^{2+}$ < 100 µM, $Ca^{2+}$ was added to solutions containing 2 mM EGTA or 2 mM HEDTA, and the final $Ca^{2+}$ concentration was confirmed with Fluo-3 or Oregon Green BAPTA 5N (ThermoFisher Scientific). For whole-cell recording, the pipette solution contained 150 mM NaCl, 10 mM HEPES, 5 mM HEDTA and 4.1 mM $CaCl_2$, and the final free $Ca^{2+}$ concentration (15 µM) was confirmed with Oregon Green BAPTA 5N. The osmolality of each solution was adjusted to 290 ~ 310 mOsm/kg.

Solutions with NaCl lower than 150 mM were made by mixing the above solutions (if $Ca^{2+}$ < 100 µM) with isotonic mannitol solution (300 mM mannitol, 10 HEPES), NMDG-MES solution (150 mM NMDG, 150 mM MES, 10 mM HEPES), or $NMDG_2$-$SO_4$ solution (150 mM NMDG, 75 mM $H_2SO_4$, 10 mM HEPES). $Ca^{2+}$ concentration was subsequently confirmed with Oregon Green BAPTA 5N. The 1 mM $Ca^{2+}$ solutions were made by directly mixing the solution containing 150 mM NaCl, 10 HEPES with the respective isotonic solutions, and $Ca^{2+}$ was added at the end. Solutions in *Figure 6F* contained 150 mM NaI, Na-gluconate or NMDG-Cl with 10 mM HEPES, 1 mM $Ca^{2+}$. $Zn^{2+}$ and $Gd^{3+}$ were directly added to solution of 150 mM NaCl, 10 HEPES from 100 mM stocks. The pH values of all the solutions were confirmed to be ~7.3. All the chemicals were purchased from Sigma-Aldrich (St Louis, MO).

## Electrophysiology

Cells were lifted with trypsin-EDTA (Life Technologies, Carlsbad, CA) and plated onto 12 mM coverslip (Warner Instruments, Hamden, CT) 3 ~ 4 days before recording. For recording, coverslips with cells were transferred to a chamber on a Nikon-TE2000 Inverted Scope (Nikon Instruments, Melville, NY) and transfection was confirmed with fluorescent microscopy. Patch borosilicate pipets (Sutter Instrument, Novato, CA) were pulled from a Sutter P-97 puller with resistances of 2–3 MΩ for inside-out patch recordings. Solutions were puffed to the excised patch using VC3-8xP pressurized perfusion system (ALA Science, Farmingdale, NY). Data were acquired using a Multiclamp 700B amplifier controlled by Clampex 10.2 via Digidata 1440A (Axon Instruments, Sunnyvale, CA). All experiments were performed at room temperature (22–24°C).

For measurements of reversal potentials, a 3 M KCl salt bridge was used. For experiments with 150 mM NaCl in the pipette solution, based on prediction by Clampex (Molecular Devices, Sunnyvale, CA), the liquid junction potentials for 15 mM NaCl, 45 mM NaCl (balanced with mannitol) were −1.7 mV and −1.0 mV respectively and were only corrected for calculation of permeability ratios ($P_{Na+}$/$P_{Cl-}$). For recordings with 150 mM NaCl pipette solution, the liquid junction potentials for all the other conditions were within ±2 mV, and were not corrected in the figures. For recordings where the pipette solution contained 15 mM NaCl (balanced with mannitol, NMDG-MES or $NMDG_2$-$SO_4$), the liquid junction potentials were corrected by adding 12 mV, 6 mV and 2 mV, respectively, to the measured values, as estimated by Clampex. The data in *Figure 6—figure supplement 2B*~F were displayed with the correction of liquid junction potentials.

## Data analysis

All data were analyzed using pClamp10 (RRID: SCR_011323, Molecular Devices, Sunnyvale, CA), OriginLab (OriginLab Corporation, Northampton, MA), and Graphpad Prism (RRID:SCR_002798, GraphPad Software, La Jolla, CA). For the measurement of $Ca^{2+}$-sensitivity, every trace was fitted to the Hill equation to generate its respective $EC_{50}$ and H (Hill coefficient). The curves in the figures display the averaged current magnitudes normalized to their respective maximal values ($I/I_{max}$ %). $P_{Na+}/P_{Cl-}$ values were calculated with the simplified equation derived from Goldman-Hodgkin-Katz equation:

$$\Delta E_{rev} = 59 \times \log[(P_{Na+} \times [Na^+]_o + P_{Cl-} \times [Cl^-]_i)/(P_{Na+} \times [Na^+]_i + P_{Cl-} \times [Cl^-]_o)],$$

where $[Na^+]_o$, $[Cl^-]_o$, $[Na^+]_i$, and $[Cl^-]_i$ are extracellular and intracellular $Na^+$ and $Cl^-$ concentrations, respectively.

Significant differences were determined with Student's $t$-test and ANOVA unless otherwise stated. In all cases, data represent mean ± SEM, $*p<0.05$, $**p<0.01$, $***p<0.001$, $****p<0.0001$, n.s. $p>0.05$.

## Acknowledgements

We thank Dr. Christian Peters (University of Illinois, Chicago, IL) and Dr. Huanghe Yang (Duke University, Durham, NC) for their critical reading of the manuscript and for helpful discussions. This study is supported in part by NIH Grants R01NS069229 (to LYJ), F32HD089639 (to MH) and by a grant from The Jane Coffin Childs Memorial Fund for Medical Research (to TWH). YNJ and LYJ are Howard Hughes Medical Institute investigators.

# Additional information

## Funding

| Funder | Grant reference number | Author |
| --- | --- | --- |
| National Institute of Neurological Disorders and Stroke | R01NS069229 | Lily Yeh Jan |
| Jane Coffin Childs Memorial Fund for Medical Research | | Tina W Han |
| Eunice Kennedy Shriver National Institute of Child Health and Human Development | F32HD089639 | Mu He |
| Howard Hughes Medical Institute | | Yuh Nung Jan<br>Lily Yeh Jan |

The funders had no role in study design, data collection and interpretation, or the decision to submit the work for publication.

## Author contributions

Wenlei Ye, Conceptualization, Data curation, Formal analysis, Investigation, Writing—original draft; Tina W Han, Conceptualization, Resources, Data curation, Funding acquisition, Writing—review and editing; Mu He, Conceptualization, Data curation, Funding acquisition, Writing—review and editing; Yuh Nung Jan, Resources, Supervision, Funding acquisition; Lily Yeh Jan, Conceptualization, Resources, Formal analysis, Supervision, Funding acquisition, Validation, Project administration, Writing—review and editing

## Author ORCIDs

Wenlei Ye (iD) https://orcid.org/0000-0002-4694-1493
Yuh Nung Jan (iD) http://orcid.org/0000-0003-1367-6299
Lily Yeh Jan (iD) https://orcid.org/0000-0003-3938-8498

Decision letter and Author response
Decision letter https://doi.org/10.7554/eLife.45187.024
Author response https://doi.org/10.7554/eLife.45187.025

## Additional files

### Supplementary files
• Transparent reporting form
DOI: https://doi.org/10.7554/eLife.45187.022

### Data availability
All data generated or analysed during this study are included in the manuscript and supporting files.

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
