## [Decision Letter]

[Editors’ note: this article was originally rejected after discussions between the reviewers, but the authors were invited to resubmit after an appeal against the decision.]

Thank you for submitting your work entitled "Dynamic change of electrostatic field in TMEM16F permeation pathway shifts its ion selectivity" for consideration by *eLife*. Your article has been reviewed by three peer reviewers, and the evaluation has been overseen by Kenton Swartz as the Reviewing Editor and a Senior Editor. The following individual involved in review of your submission has agreed to reveal their identity: Gilman Toombes (Reviewer #2).

Our decision has been reached after consultation between the reviewers. Based on these discussions and the individual reviews below, we regret to inform you that your work will not be considered further for publication in *eLife*.

The reviewing editor and reviewers found your manuscript on TMEM16F to be interesting and provocative. However, after careful consideration, the consensus was that there are substantive technical concerns with your measurements of V_rev_, including potential problems arising from not subtracting background conductances or controlling for potential changes in ion concentrations during your recordings. There were also concerns with your proposed mechanism and whether the changes you see may be physiologically relevant, The reviewers articulate their concerns at considerable length, which we hope you will find helpful in sorting out whether the ion selectivity of TMEM16F is indeed dynamic, and if so what the underlying mechanism and physiological relevance may be.

*Reviewer #1:*

This manuscript by the Jan lab investigates the selectivity of the TMEM16F channel/scramblase. Multiple contrasting publications in the literature reported that TMEM16F-mediated ionic currents are anion-, cation- or non-selective. Here, Ye and colleagues show that the selectivity of TMEM16F changes dynamically as a function of divalent cations, ionic strength, and voltage. The authors propose that this shift in selectivity is not due to the ability of the TMEM16F ion pore to adopt conformations with different selectivity properties, but rather to the modulation by divalent cations of the electric field within the pore.

These findings are important because they provide a unifying framework to interpret the contrasting and confusing data present in the literature. However, I have three major concerns with regards to the interpretation of the data, and several minor issues that need to be addressed.

1) TMEM16F is a scramblase, as such its main function is lipid transport to externalize PS to initiate blood coagulation and recognition of apoptotic cells by macrophages. The physiological relevance of TMEM16F-mediated ionic currents is -at the moment- unclear. Further, activation of TMEM16F currents require sustained depolarizations and high Ca^2+^ concentrations, while lipid scrambling requires only elevation of Ca^2+^, but not depolarization. Thus, it is not clear under what physiological conditions the self-braking mechanism enabled by the dynamic switch in selectivity might play a role. For example, does the membrane potential of platelets and/or apoptotic cells reach and maintain the depolarizations necessary for channel activation? Furthermore, the magnitude of the observed switch is relatively modest; in all conditions the channel is poorly selective between anions and cations, displaying only modest preferences for one over the other. In no cases do the currents approach ideal anion vs cation selectivity. Given the overall poor selectivity, I wonder how much of an effect would be caused by the observed switch in selectivity. Thus, while the phenomenon described here is biophysically interesting, I am not convinced of its physiological relevance. Some quantitation of the effects observed here within the context of a cell would be needed to establish a clear link to physiology.

2) It is not clear through what type of pore ions move in TMEM16F. It has been proposed that the TMEM16 scramblases form a lipid-lined pore through which ions can diffuse together with the lipids that are being scrambled (Whitlock and Hartzell, 2016; Jiang et al., 2017). If this were the case, then the selectivity of the channel would be affected by multiple factors such as lipid headgroups, voltage and mono- and di-valent ions. On the other hand, the TMEM16A homologue forms a more conventional, protein-delimited, pore. In this case, the observed changes in selectivity could reflect either a charge-screening effect (as the authors propose) or the ability of the channel's pore to adopt slightly different conformations. Further, most of the measurements are carried out using a mutant, Q559K that affects both selectivity and gating of TMEM16F. I understand that the mutant was used because of the removal of run-down. It is unclear whether this construct can serve as a good model for the WT pore. The mutation could case subtle rearrangements in the pore rendering it more sensitive to environmental changes. While the present experiments provide evidence suggesting that the selectivity of TMEM16F changes, they shed little light into the mechanisms underlying such a switch.

3) A technical concern I have is that the measured currents are relatively large (~1 nA) and the stimulation protocols used are long. Can the authors rule out that the effects they see are due to ion accumulation/depletion effects? The currents in 15 μm Ca^2+^ are naturally smaller than those in 1 mM Ca^2+^, and therefore these effects are less pronounced leading to the observed differences. Did the authors use different directs of the ramp and/or tail current protocols to measure reversal potentials? Is there an effect on the direction of the ion switch (i.e. going from 150 NaCl to 15 NaCl is the same as going from 15 NaCl to 150 NaCl)? Do they observe a dependence of the magnitude of the effects as a function of the current in the patch?

4) Subsection “Q559K channel ion selectivity varies with intracellular Ca^2+^ concentration”. The reversal potential of the QK/EQ mutant suggest this mutant is less Cl^-^ selective than the QK mutant alone. Thus, at an intermediate Po the selectivity is also intermediate. This is more consistent with a model where the pore can adopt multiple open states with different selectivity. What happens to the currents of the QK mutant alone (or WT) when measured at a Po~0.5? I do not think that the authors' conclusion that "This indicates either that TMEM16F ion selectivity is not correlated with its open state, or that the E667Q mutation circumvents the intermediate open state (if any) and causes the mutant channel to directly enter a fully-open conformation" is warranted by the data.

5) Subsection “The switch of ion selectivity correlates with electrostatic change in permeating pathway”. If the shift in V_rev_ depends on divalent ions (Ca^2+^, Zn^2+^) then the shift should depend in a graded manner on their concentration. Did the authors test what happens when using intermediate increases between the 15 and 1000 μm concentrations shown?

6) Are divalent anions inert? Since the channel is permeable to anions under all conditions (even when it is in its "cation selective" state, its selectivity is modest), these ions can enter the pore and therefore should have the opposite effect of the divalent cations.

7) The inactivation in Figure 4A is slow (tau~30s – 1 min). Why weren't the reversal potentials measured after reaching activation but before inactivation sets in? Is inactivation dependent on voltage? If the patches are held at a membrane voltage closer to reversal is there a similar effect?

8) I find the effects of Zn^2+^ confusing where they regard the rapid run-down that is PIP_2_ independent. Are there multiple run-down mechanisms?

9) Discussion section. The authors state "However, TMEM16F does not have an "Ohmic" state: the channel always requires depolarization to be activated." What does this mean? That the pore is intrinsically rectifying? Ohmic state would refer to conductance, not gating.

*Reviewer #2:*

While TMEM16F's dual roles as a calcium-activated phospholipid scramblase and ion channel are well established, its’ ionic selectivity remains the subject of debate. In this work, the authors examine whether the ionic selectivity of TMEM16F can change in response to solution composition and membrane voltage. Using the Q559K mutation to prevent channel rundown, the authors report that TMEM16F switches its preference from cations to anions depending on the intra-cellular calcium concentration and membrane voltage used to activate it, and they propose that this switch results from the calcium ions and membrane potential changing the electrostatic field in the permeation pathway.

The authors results are very interesting, but I have several concerns.

1) Experimental artifacts: To combat channel rundown and voltage-dependent activation, the authors have resorted to protocols with sizeable resting membrane currents and conductance, conditions that can easily lead to ion accumulation/depletion artifacts ("Calcium-calmodulin does not alter the anion permeability of the mouse TMEM16A calcium-activated chloride channel", JGP 2014 Jul; 144(1): 115-124.).

I am sure the authors have thought carefully about this issue and probably already have data to exclude this possibility. For example, since the channel undergoes desensitization, they should have recordings in which early ramps have a much larger conductance than later ramps recorded under identical conditions (e.g. [Ca^2+^], holding voltage). If the reversal potential for each condition is essentially independent of the holding current/conductance, ion accumulation/depletion artifacts are unlikely. Reporting key experimental details (e.g. complete protocol and order in which conditions/ramps were performed, leak subtraction protocol, series resistance compensation, composition of 15mM NaCl solution, control experiments to exclude effects of Ca^2+^, HEPES, EDTA permation, etc.) would help address these concerns.

2) Proposed mechanisms for holding voltage changing ionic selectivity:

The authors seem to be proposing that Ca^2+^ (and perhaps also Na^+^) ions driven into the permeation pathway by the electric fields/driving force during the holding/activation voltage somehow modify permeation during the subsequent voltage ramp step. I have two concerns with this idea. Firstly, the authors have previously reported that TMEM16F is quite permeable to calcium. Larger depolarizations will drive more calcium out through the channel which could lower the calcium concentration on the intra-cellular side of the permeation pathway, rather than increasing it. Secondly, for the permeation pathway to act as a channel, ions must be able to enter and leave it on the microsecond timescale and thus one would expect rapid turnover of all permeating ions during the voltage ramp. For the channel to "remember" the holding voltage, it seems like voltage would either have to induce a conformational change or modulate the binding of ions to specific sites that turnover slowly enough to remain present during the subsequent ramp.

*Reviewer #3:*

This manuscript explores how Ca^2+^ concentration may affect TMEM16F ion-selectivity. In high intracellular Ca^2+^, TMEM16F runs down quickly. However, the authors overcome the rundown problem of TMEM16F by utilizing the Q559K mutant together with the wild-type TMEM16F for their experiments. In both proteins, they report that the current is cation selective at low intracellular [Ca^2+^], but becomes anion selective at high Ca^2+^. Also, they show that manipulating the opening state of the channel does not affect this ion-selectivity shift, but manipulating electrostatic field in the permeation pathway of the channel does. Finally, they also found that more positive potential tends to increase the anion permeability. They speculate that the discrepancy of TMEM16F ion selectivity reported in the literature may result from different recording conditions such as holding potentials and ion concentrations.

While resolving the controversy in the cation/anion selectivity of TMEM16F is important, the authors' conclusion that the controversy results from different recording conditions used in various labs (such as different holding voltages, ionic conditions, or Ca^2+^ concentrations used) is a little bit hand waiving. In the literature, the permeability difference in TMEM16F between Na^+^ and Cl^-^ is not huge. Even accepting the results in this study as face value, the TMEM16F's P_Na_/P_Cl_ ratios range from 0.5 to 2 (Figure 5D). It is therefore strange that the authors would call such a several-fold difference of Na^+^/Cl^-^ permeability as Na^+^ selective (or Cl^-^ selective in the other way around). Even worse, the changes of the P_Na_/P_Cl_ permeability ratios reported here may actually be artefacts.

In the present study and the Yang et al. paper cited by the authors (Yang et al., 2012), there are at least two serious technical problems that could contribute to the controversy. In fact, the result from Yang et al., 2012 and the result in this paper, both from the same lab, already contradict each other. Here is a striking example: in the present study, the authors show that with a high concentration (1 mM) of Ca^2+^, when [NaCl]i is changed from 150 mM to 15 mM, the reversal potentials (in both Q559K and wild-type TMEM16F) shift from ~ 0 mV to ~ -15 mV (Figure 1). In Yang et al., 2012, a similar 10-fold change of [NaCl]i in the presence of 0.5 mM Ca^2+^ generated a right shift of the reversal potential to ~+20 mV (Figure 6 D and 6G in Yang et al., 2012). Both experiments are excised inside-out patch recordings, and the solutions are similar (150 mM/15 mM vs 140 mM/14 mM). The [Ca^2+^]i in Yang et al. was 0.5 mM, which should be closer to 1 mM Ca^2+^ than to 15 µM Ca^2+^. Why did the authors obtain a left shift of the reversal potential while in Yang et al., 2012, the reversal potential shifted to the right?

Two technical issues in both the current study and Yang et al., 2012 are not seriously dealt with.

The first serious technical problem is the junction potential correction. Throughout the whole paper of Yang et al., 2012, there was no description of junction potential correction. In the present study, the authors mention that "the liquid junction potentials for 15 mM NaCl, (and) 45 mM NaCl were -1.7 mV and -1.0 mV respectively.……". I suppose that the authors calculated these values with respect to the 150 mM NaCl solution. If so, these values are severely under-estimated. In fact, using the same calculator (Clamplex), the value of liquid junction potential for 15 mM NaCl (with respect to 150 mM NaCl) is -11 mV (I confirm this value by actually measuring the liquid junction potential between 15 mM NaCl and 150 mM NaCl solution). Since the data points in the figures (For example, Figure 1 F and G) are not corrected for the junction potential, the reversal potential of ~ -15 mV in 1 mM Ca^2+^ (see Figure 1G) should only be ~ -4 mV (with a 10-fold NaCl gradient). The reversal potential shift in Q559K in 15 µM Ca^2+^ in the paper is ~ +10 mV (Figure 1G). After junction potential correction, this should be ~+20 mV.

The second and even more problematic issue is a lack of background current subtraction. Throughout the whole manuscript (and also in Yang et al., 2012), I do not see a description of background current subtraction for the presented I-V curves (and thus the measured reversal potentials). In the absence of TMEM16F conductance, the HEK293 cells have a cation-selective endogenous conductance. The authors actually show this fact from the background current measurement where the reversal potentials under 150 mM/15 mM NaCl condition were ~+20 mV (in 1 mM Ca^2+^) and ~ +60 mV (in 15 µM Ca^2+^) (see B and C in Figure 1—figure supplement 1). The conductance of the untransfected cell patches includes the leak conductance and the conductance from the endogenous channels/transporters. If we sum the two as the background conductance (abbreviated as gb), and the conductance from TMEM16F (the channel conductance) is denoted as gc, the measured reversal potential (E_rev_) should be a weighted sum of the reversal potential of gc (abbreviated as Ec) and that of gb (abbreviated as Eb) according to the following equation:

E_rev_ = [gc/(gc+gb)] × Ec + [gb/(gc+gb)] × Eb

Reducing [NaCl] or Ca^2+^ will reduce the TMEM16F conductance (gc), and therefore, E_rev_ will approach Eb (which is +20 mV or +60 mV). When the gc is much larger than gb (this is the case when TMEM16F is activated by 1 mM Ca^2+^), the contribution of gb to the measured E_rev_ is less significant (as shown in Figure 1—figure supplement 1A, blue and grey curve). However, when gc is small relative to gb, the measured E_rev_ will be severely contaminated by Eb. The example is no better than the one provided by the authors in Figure 6, where when the holding voltage is more positive (the TMEM16F conductance is larger because of its outward rectification property), the reversal potential is less positive (less contaminated by Eb). This phenomenon is consistent with all the results throughout the paper. Various manipulations in the paper (including several mutations or even experiments in TMEM16A) would not change this trend-the larger the gc, the less positive the measured E_rev_.

Without subtracting the background current obtained in the absence of Ca^2+^, how large the total current should be so that the measured E_rev_ is good enough to reflect the true TMEM16F reversal potential (Ec)? The equation above indicates that gc needs to be larger than 90% of the total conductance (gc + gb) in order to have a measurement where <10% of Eb (namely +2 mV to +6 mV, according to the measured Eb) is added to the measured reversal potential. The authors show that the background current from untransfected cell patches at +80 mV is 50-80 pA (Figure 1—figure supplement 1B and C). If the total current at +80 mV is 500 pA (0.5 nA) in the solution with 15 µM Ca^2+^ and 15 mM NaCl, as in many of the authors' recording traces, the conductance of TMEM16F at +80 mV is ~ 9-10 fold that of the background current. However, since WT TMEM16F and Q559K mutant are outwardly rectifying, their conductances are quickly reduced by several fold near the reversal potential (the outward rectification is even more severe in low NaCl and low Ca^2+^ condition). Therefore, the 9-10 fold gc/gb ratio at +80 mV may reduce to 2-3 near Ec (gb is constant at different voltages). Suppose the true value of Ec is 0 mV in the 150 mM/15 mM NaCl condition (namely, P_Na_/P_Cl_ =1), and the value of Eb is +60 mV (as shown in the supplementary figure), a gc/gb ratio of 2-3 will give a measured reversal potential of +20 mV – +30 mV. If the total current at +80 mV is less than 0.5 nA, the situation will be even worse. Interestingly, the reversal potentials measured in 150 mM/15 mM NaCl condition in many experiments in this manuscript are in this range.

The above example therefore emphasizes the importance of subtracting the background current in order to evaluate EC more precisely. However, I am only using the data provided by the authors to make the above estimate. Since the background currents vary from patch to patch, the easy way to minimize the contribution of the background conductance is to subtract the current obtained in the absence of Ca^2+^ from current obtained in the presence of Ca^2+^. By doing background current subtraction, the authors would eliminate the shift in E_rev_ caused by the background conductance and thus isolate the shift in E_rev_ caused by Ca^2+^. It is interesting that similar background current contamination of the reversal potential measurement is also shown in TMEM16A and its K584Q mutant in this manuscript (Figure 6—figure supplement 1). However, it appears that the authors have not noticed this problem. Taking K584Q recordings as an example (see Figure 6—figure supplement 1), the K584Q conductance is bigger in 1 mM Ca^2+^ than in 0.4 µM Ca^2+^. Therefore, the reversal potential is more shifted to the right in 0.4 µM Ca^2+^. Here, the measured reversal potentials (10 fold NaCl gradient) of K584Q in 0.4 µM and 1 mM Ca^2+^ are both far from the perfect Cl^-^ Nernst potential. It has already been shown (see Supplementary figure 2 in Jeng et al., 2016) that after background current subtraction, the K584Q mutant, like the wild-type TMEM16A, is perfectly Cl^-^ selective in 20 µM Ca^2+^. In other words, based on the measured reversal potentials, K584Q in 20 µM Ca^2+^ in Jeng et al., 2016, is more Cl^-^ selective than the K584Q current activated by 1 mM Ca^2+^ shown in this paper. The key difference between the two studies is the background current subtraction.

With these technical flaws, the reported reversal potentials in this manuscript are simply not correct. I am not sure if the authors have measured background I-V curve in every recorded patch. If not, I do not think the problem can be corrected within a short time. If these two technical problems are corrected, I anticipate that conclusion of the paper is probably not going to hold.

[Editors’ note: below is the decision letter following the authors’ appeal.]

Thank you for choosing to send your work entitled "Dynamic change of electrostatic field in TMEM16F permeation pathway shifts its ion selectivity" for consideration at *eLife*. Your letter of appeal has been considered by Rick Aldrich as Senior Editor and Kenton Swartz as Reviewing editor, and we are prepared to consider a revised submission with no guarantees of acceptance.

The editors and reviewers have read and discussed your rebuttal. It is important to point out that our initial concerns remain, and we are not convinced that your plan for revision will satisfy those concerns. However, we are willing to reconsider a revised manuscript after you have had a chance to do the additional experiments. The following are several points that you should consider in preparing the revised manuscript.

1) We remain concerned that leak currents could contribute in a non-linear fashion to the observed shifts in V_rev_. The data provided in your rebuttal does not address this issue, as in different ionic concentrations the shape and V_rev_of the leak will change. We suggest that you bracket each IV measurement in any solution containing calcium with measurements of the background IV (i.e. 0µM Ca^2+^) in solutions with identical NaCl concentration so that the voltage dependence of the total and leak currents can be directly compared. We understand that there is no perfect way to identify and subtract leak currents for TMEM16F, but by showing raw and leak corrected data it will be possible to estimate the total and leak conductances at the reversal potential and thereby gauge the accuracy and reliability of the V_rev_calculations. Since this does not appear to have been done in your initial experiments, we would request that they show a set of experiments between the extremes of low and high current density where you do these experiments, perhaps using a couple of different constructs.

2) As you point out, the current in your experiments is typically quite high, in the range of 1-8 nA when activated with calcium at +80mV. Although increasing current density diminishes the contribution of leak currents, currents of this magnitude will cause substantial voltage errors due to series resistance and will also alter ion concentrations (via ion depletion and accumulation). The plots of the current magnitude vs V_rev_shift in Author Response Image 1D of your rebuttal only partially addresses this concern since the conductance of TMEM16F is nonlinear and you have not reported the TMEM16F and leak conductances at the reversal potential. Furthermore, the data in Author Response Image 1D shows that there is a ~2-fold variability in the V_rev_measurements (between ~15 and ~35 mV), in a similar range to the changes in V_rev_that lead you to conclude that the ion selectivity of TMEM16F changes. We are not certain how best to resolve this issue, but we suggest that it might be helpful to provide V_rev_measurements as outlined above for both low and high calcium and to extend the range shown in Figure 1D to somewhat lower current density, in the range of 500 pA. While these might be affected more by background leak currents, they might also serve to validate the conclusions by providing an empirical calibration of the role of the leaks and other errors.

3) We suggest that you rephrase how you describe the ion selectivity of TMEM16F. Assuming a clear shift in V_rev_can be unambiguously demonstrated, it would be more appropriate to refer to the protein as a poor ion selective channel that slightly changes its ionic preference (rather than a sodium selective channel that becomes chloride selective).

4) We think the potential physiological effects of the ion selectivity changes you report are overstated. You argue that the effects might induce a ~13 mV change in the resting potential of a cell where TMEM16F is the main conductance. However, we are unaware of conditions where a cell would be exposed to millimolar intracellular calcium and sustained large depolarizations. Moreover, the presence of other currents would significantly diminish the impact of small changes in the selectivity of TMEM16F, especially since in the -80 to +50 mV range (where the cell membrane potential is likely to be), TMEM16F has a very low Po (i.e. Figure 1—figure supplement 2D). Therefore, the physiological effects on a cell's membrane voltage -if any- will be considerably smaller than ~13 mV. We request that the physiological relevance should be removed from the Abstract and significantly toned down in the Discussion. The focus of the manuscript should be on the biophysical mechanism underlying the small shift in selectivity.

5) We suggest that you add a discussion about the origin of the discrepancy of the ion selectivity of TMEM16A and TMEM16F.

[Editors' note: further revisions were requested prior to acceptance, as described below.]

Thank you for resubmitting your work entitled "Dynamic change of electrostatic field in TMEM16F permeation pathway shifts its ion selectivity" for further consideration at *eLife*. Your revised article has been favorably evaluated by Richard Aldrich (Senior Editor), Kenton Swartz (Reviewing Editor), and three reviewers.

The manuscript has been improved but there are some remaining issues that need to be addressed before acceptance, as outlined below:

Summary:

The authors' extensive revisions resolve many issues with the earlier version. The article now describes the authors' considerable efforts to exclude artifacts that might bias the reversal potential methods, and also provides sufficient detail that their measurements can be meaningfully compared to other published work. The following are suggestions for revising the manuscript:

Major comments:

1) The term "electrostatic gate" seems out of place. The ordinary meaning of a "gate" is to control a flux in an all-or-none way. It is very odd to read a sentence like “…the "closed state" of the electrostatic gate allows the permeation of cations.”. Throughout the whole paper, there are no results showing that TMEM16F is only conducting cations or only conducting anions. It seems inappropriate to use the concept of "gate" to represent the control by a change of electrostatic potential. Controlling the cation versus anion selectivity by altering the pore potential (through charge residue mutations) is well documented in pentameric ligand-gated channels. The authors may want to reference those papers to strengthen the main conclusion in this paper.

2) Synergy of Ca^2+^ and Depolarization: The authors propose that depolarization indirectly modifies permeation by "driving Ca^2+^ into the membrane electric field". While the conceptual simplicity of this idea is appealing, the authors observe that depolarization still alters the relative permeability even at high (i.e. 1mM) calcium concentrations where the site of action for calcium should be saturated.

---

## [Author Response]

[Editors’ note: the author responses to the first round of peer review follow.]

We thank the reviewers for their helpful suggestions. We are grateful for the acknowledgement by all the reviewers that our work provides a unifying mechanism for previous discrepancy pertinent to ion selectivity of TMEM16F. Parts of comments by reviewer 1 and 2, and almost all of the comments by reviewer 3 are focusing on technical issues, mainly questioning whether the shift of reversal potential is an artifact resulting from issues involving ion accumulation / depletion, liquid junction potential and background current contamination. Reviewers 1 and 3 also raise the question about the physiological implication of the ion selectivity shift. In addition, reviewers 1 and 2 inspire us to perform a few more practical experiments to support our proposed mechanism. In this appeal letter, we will clarify these main issues followed by point-to-point responses.

We will address our responses to the concerns raised by multiple reviewers, including technical issues and physiological implications, and believe these concerns could be cleared. Upon the request by the reviewers, we will perform a few additional experiments to corroborate our mechanistic model.

1) Technical issues

All the reviewers express the concern regarding whether the shift of E_rev_ is an experiment artifact. We apologize for not having explained the methodology clearly in the manuscript. We will add the experimental details and two figures (Figure 1—figure supplement 2 and Author response image 1) to the revised manuscript to illustrate how the traces were obtained from recording with a hyperpolarizing ramp, and how we performed “quality control”, as suggested by reviewer 2. We hope this revision will address the reviewers' concerns.

First, for all the experiments, we started recording in a 150 mM NaCl, Ca^2+^-free solution and made sure that the current recorded at +80 mV is around or smaller than 100 pA before applying Ca^2+^ (Figure 1—figure supplement 2 A, G); In most cases we only included recordings with Ca^2+^-activated current > 1000 pA. In all cases, we made sure that the Ca^2+^-activated current (recorded at +80 mV) is >ten-fold larger than endogenous current before switching protocol for testing E_rev_, and this is consistent with reviewer 3's estimation. In addition, for data in Figure 1 (and supplement 2, 4), Figure 2, Figure 3, Figure 4, Figure 5, Figure 6—figure supplement 1, Figure 7, we always recorded the current in 0 Ca^2+^ solutions at the end to make sure that the seal is good (as in Figure 1—figure supplement 2 B, E, H, K).

Second, there is no correlation between reversal potential and current magnitude (Figure 1—figure supplement 2 D, J). If the current was contaminated by endogenous current (which is cation-selective), there would be a negative correlation between E_rev_ and current magnitude, but we did not see such a trend. This analysis also partially answers the question of reviewer 1 and 2 regarding artifacts caused by ion accumulation / depletion.

Third, we did notice that with the hyperpolarizing ramp (so-called "reverse ramp"), using current magnitude at +80 mV as “quality control” will underestimate the effect of background current contamination, because TMEM16F current is outwardly rectifying while background current is linear (as pointed out by reviewer 3). We have thought about this and have made sure that our conclusion is valid: (1) In Figure 1—figure supplement 2C, the current at 0 mV is ~-200 pA, which would not have been "eliminated" even if we performed background subtraction, suggesting that the channel activated by 15 µM Ca^2+^ mostly permeates cation. (2) The "reverse tail current" (recorded at +80 mV, arrowed in Figure 1—figure supplement 2C, I, L) following the reverse ramp is still 6~7-fold the size of the background current, meaning that the current at ~0 mV will be larger than that. The current underlying the reversal potential thus is dominated by TMEM16F current. To be cautious with our conclusions, we did not interpret the reversal potential of WT TMEM16F in 1 mM Ca^2+^ (Figure 1—figure supplement 2F). (3) Most importantly, the double mutant Q559K_G615A reduces the rectification in both 15 µM and 1 mM Ca^2+^(Author response image 1). For this double mutant, the currents recorded at both Ca^2+^ concentrations were almost linear (Figure 4E, Author response image 1) with little inactivation at hyperpolarized membrane potential, and the reversal potential was clearly shifted.

Fourth, reviewer 1 and 2 raise the issue of ion accumulation / depletion. For most recordings, we switched solutions "back and forth" to make sure traces in the same solution reversed at the same voltage despite the change of current magnitude. We also successively recorded multiple traces in the same solution to make sure there was no drift. Between different cells, there was no correlation between E_rev_ and current magnitude (Figure 1—figure supplement 2). In addition, the main conclusion of our study is not affected by any deviation (even if there is actually a deviation) caused by the ion accumulation or depletion: We used a hyperpolarizing ramp protocol following a holing potential at +80 mV (See Figure 1—figure supplement 2). This will cause Na^+^ efflux and/or Cl^-^ influx, potentially leading to Na^+^ accumulation on the extracellular side and Cl^-^ accumulation on the intracellular side (or intracellular Na^+^ depletion and extracellular Cl^-^ depletion). All of these possibilities should push ENa and ECl to “more positive” values, leading to a positive shift of E_rev_ (as observed in the JGP paper1 cited by reviewer 2). This is opposite to the difference between 15 µM and 1 mM Ca^2+^, where the E_rev_ in 1 mM Ca^2+^ is more negative. Also, In Figure 6D, a more depolarized holding potential should exacerbate the accumulation/depletion effect (if there is any), also pushing E_rev_ to more positive values. The data show the opposite.

Fifth, reviewer 3 states that "the smaller the magnitude, the more cation-selective the current" suggesting that the current is contaminated by endogenous current. We believe this is a misinterpretation of our data. The reviewer might be comparing the magnitudes between the currents in 15 mM NaCl (~0.5 nA, as was stated in his/her opinion, likely referring to the red and blue traces in Figure 1F), which seems inappropriate; instead, we should compare the pink trace and the gray trace (Figure 1F, recorded in 150 mM NaCl). For each cell expressing Q559K mutant channels, the current activated by 15 µM Ca^2+^ (pink) and by 1 mM Ca^2+^ (gray) were similar in magnitude (consistent with data in Figure 1B, D). In contrast, since the red trace is more Na^+^-selective than the blue trace, in 15 mM [NaCl]_i_ the driving force for Na^+^ efflux at +80 mV is small (only 20 mV), while in 1 mM Ca^2+^ (the blue trace) the outward current is mainly carried by Cl^-^ influx. Thus, this comparison between the red and blue traces is in fact consistent with our main idea. In addition, among different cells expressing WT or Q559K channels, there is no correlation between current magnitude and reversal potential (Figure 1—figure supplement 2 D, J).

Last but not least, reviewer 3 raises the issue of background current contamination and liquid junction potential. Given the above explanation of experiment details, including selection of Ca^2+^-activated current > 1 nA, validation with "reversal tail current" and the utilization of a mutant with reduced rectification, we hope the reviewer would be convinced that the measured reversal potentials are accurate. Also, in the point-to-point response to reviewer 3, we will demonstrate that background current correction will actually "expand" the magnitude of E_rev_ shift and cause additional problems, given how Ca^2+^ affects the background current. We will also demonstrate in our response to reviewer 3 that our correction of liquid junction potential is reliable.

2) Physiological implication

Reviewer 1 and 3 point out that the shift of reversal potential is modest and question whether it is physiologically significant. We believe that the “relatively modest” ion selectivity change is sufficiently large to be relevant for physiological functions. In a hypothetical cell whose Cl^-^ equilibrium potential is -60 mV and extracellular / intracellular cations are Na^+^ / K^+^ respectively, a shift of P_Na+_/P_Cl-_ from 0.5 to 2 can cause the membrane potential shift from -22 mV to -9 mV (assuming that this channel is the predominant local membrane conductance, and its P_Na+_ = P_K+_), big enough to regulate some signaling molecules such as NMDA receptors, voltage-gated Ca^2+^ and Na^+^ channels. Given that TMEM16F is proposed to be on the dendritic spines of neurons^2^, a local membrane potential change can be very consequential.

TMEM16F is one of the several TMEM16 members universally expressed in diverse cell types, including epithelial cells, neurons and immune cells, but its physiological functions have not been fully characterized. It was reported that TMEM16F is localized in the primary cilia of epithelial cells^3^ as well as odorant receptor neuron sensory cilia^4^. Primary cilia are specialized organelles containing high Ca^2+^ and being constantly depolarized^5^, offering a “platform” for the dynamic ion selectivity. The ion transport in primary cilia is still a new area of investigation, and our work is indicative of a new regulatory machinery of primary cilia signaling. Our studies of TMEM16F might provide novel hints to people studying ion transport at subcellular regions, such as dendritic spines and cilia.

3) Additional experiments as suggested by reviewers

Based on suggestions of reviewer 1, we will test the effect of divalent anion on reversal potential. We will use Na_2_SO_4_-based pipette solution and record the reversal potential of Q559K with the same protocol. We expect to see a less substantial reversal potential shift as we switch from 15 µM Ca^2+^ to 1 mM Ca^2+^, since we predict divalent anions might have effects opposite to those of divalent cations with respect to the alteration of electrostatic field in the permeation pathway.

Based on the suggestions of reviewer 2, we will test TMEM16F's "memory" of factors impacting the reversal potential at a given holding voltage (+80 mV). We will hold the membrane potential at +80 mV for ~5 s followed by hyperpolarizing ramps of different speeds. We expect to see that the recorded reversal potential is shifted to a more positive value as the ramp speed is reduced. This would indicate that TMEM16F ion selectivity is restored to being more Na^+^ selective as the membrane repolarizes during a slow ramp.

Also based on the suggestion of reviewer 2, we will test the reversal potential of Q559K with whole cell configuration.

Based on the comment of reviewer 3, we will test whether replacing NaCl with mannitol or NMDG_2_-SO_4_ would affect the ion selectivity of TMEM16A.

**Author response image 1. respfig1:** The shift of E_rev_ persists in a mutant with reduced outward rectification, Q559K_G615A. (**A**) Time course of TMEM16F Q559K_G615A activation by 15 µM Ca^2+^ in 150 mM NaCl. The inside-out membrane patch was held at +80 mV. (**B**) Normalized conductance-voltage relationships. (**C**) Recordings of reversal potential with a hyperpolarizing ramp protocol following constant holding potential at +80 mV. The solutions were applied in the order of 150 mM with 15 µM Ca^2+^, 15 mM with 15 µM Ca^2+^, 150 mM NaCl with 1 mM Ca^2+^, 15 mM NaCl with 1 mM Ca^2+^. Ca^2+^-free 150 mM NaCl solution was applied at the end to confirm the seal. (**D**) Detail amplification showing the tail currents in C recorded at +80 mV following the hyperpolarizing ramp.

Reviewer #1:[…] 1) TMEM16F is a scramblase, as such its main function is lipid transport to externalize PS to initiate blood coagulation and recognition of apoptotic cells by macrophages. The physiological relevance of TMEM16F-mediated ionic currents is -at the moment- unclear. Further, activation of TMEM16F currents require sustained depolarizations and high Ca^2+^ concentrations, while lipid scrambling requires only elevation of Ca^2+^, but not depolarization. Thus, it is not clear under what physiological conditions the self-braking mechanism enabled by the dynamic switch in selectivity might play a role. For example, does the membrane potential of platelets and/or apoptotic cells reach and maintain the depolarizations necessary for channel activation? Furthermore, the magnitude of the observed switch is relatively modest; in all conditions the channel is poorly selective between anions and cations, displaying only modest preferences for one over the other. In no cases do the currents approach ideal anion vs cation selectivity. Given the overall poor selectivity, I wonder how much of an effect would be caused by the observed switch in selectivity. Thus, while the phenomenon described here is biophysically interesting, I am not convinced of its physiological relevance. Some quantitation of the effects observed here within the context of a cell would be needed to establish a clear link to physiology.

We thank the reviewer for raising the question regarding the physiological relevance of our work. It is true that previous studies regarding TMEM16F functions are mostly focused on its function of phospholipid scrambling and vesiculation. However, recently published papers^6^ and preprints^7^ indicated that monovalent ions (not only Ca^2+^) are necessary for the execution of these functions. Clarification of TMEM16F ion selectivity will help unravel the mechanism of lipid scrambling, providing answers to questions such as whether scrambling occurs at a particular stage of the ion permeability transition, and whether movement of different ions drives the scrambling of different lipids.

TMEM16F is one of the few TMEM16 members universally expressed in diverse cell types, but its subcellular localization and physiological functions have not been fully characterized. It was reported that TMEM16F is localized in the primary cilia of epithelial cells^3^, which are specialized organelles containing high Ca^2+^ and being constantly depolarized^5^ – physiological conditions where TMEM16F ion selectivity shift is very likely. Our studies might suggest implications for researchers studying these subcellular regions. Also, as we discussed in "responses to all reviewers", the “relatively modest” ion selectivity change is sufficiently large to be relevant for physiological functions, such as the regulation of NMDA receptors and voltage-gated ion channels. Given that TMEM16F is proposed to be on the dendritic spines of neurons^2^, a local membrane potential change can be very consequential.

2) It is not clear through what type of pore ions move in TMEM16F. It has been proposed that the TMEM16 scramblases form a lipid-lined pore through which ions can diffuse together with the lipids that are being scrambled (Whitlock and Hartzell, 2016; Jiang et al., 2017). If this were the case, then the selectivity of the channel would be affected by multiple factors such as lipid headgroups, voltage and mono- and di-valent ions. On the other hand, the TMEM16A homologue forms a more conventional, protein-delimited, pore. In this case, the observed changes in selectivity could reflect either a charge-screening effect (as the authors propose) or the ability of the channel's pore to adopt slightly different conformations. Further, most of the measurements are carried out using a mutant, Q559K, that affects both selectivity and gating of TMEM16F. I understand that the mutant was used because of the removal of run-down. It is unclear whether this construct can serve as a good model for the WT pore. The mutation could case subtle rearrangements in the pore rendering it more sensitive to environmental changes. While the present experiments provide evidence suggesting that the selectivity of TMEM16F changes, they shed little light into the mechanisms underlying such a switch.

We thank the reviewer for the question regarding the structure of TMEM16F. Previous studies have suggested that the Q559K mutant is deficient in scrambling, suggesting that it is a mutant stabilized in an open channel conformation with scrambling defect^8^. A recent paper also proposes that mutation of Q559 establishes the feasibility to study ion transportation of TMEM16F^9^. Although WT is not feasible for mechanistic research about the impact of high internal Ca^2+^ concentration on ion selectivity, we showed that the critical phenotypes as found in Q559K mutant channel could all be recapitulated in WT, including the shift of E_rev_ in accordance to ionic strength (Figure 6), voltage (Figure 7) and Ca^2+^ level (Figure 1-figure supplement 4). Several studies of the TMEM16 family of proteins regulated in multiple ways have exemplified the strategy of using mutants to “stabilize" some dynamic features (e.g., using G644P to stabilize the TM6 conformational change of TMEM16A^10^, using Q645A to circumvent one Ca^2+^-binding event^11^), as long as critical phenotypes inferred from the mutant studies can be recapitulated in WT.

3) A technical concern I have is that the measured currents are relatively large (~1 nA) and the stimulation protocols used are long. Can the authors rule out that the effects they see are due to ion accumulation/depletion effects? The currents in 15 μm Ca^2+^ are naturally smaller than those in 1 mM Ca^2+^, and therefore these effects are less pronounced leading to the observed differences. Did the authors use different directs of the ramp and/or tail current protocols to measure reversal potentials? Is there an effect on the direction of the ion switch (i.e. going from 150 NaCl to 15 NaCl is the same as going from 15 NaCl to 150 NaCl)? Do they observe a dependence of the magnitude of the effects as a function of the current in the patch?

We thank the reviewer for the helpful suggestion. The effects we observe are unlikely due to ion accumulation/depletion. We used a hyperpolarizing ramp following holding the membrane potential at +80 mV (See Figure 1—figure supplement 2). This will cause Na^+^ efflux and/or Cl^-^ influx, supposedly leading to Na^+^ accumulation on the extracellular side of the channel and Cl^-^ accumulation on the intracellular side (or intracellular Na^+^ depletion and extracellular Cl^-^ depletion). All of these should push E_Na_ and E_Cl_ to “more positive” values, leading to a positive shift of E_rev_. This is opposite to the shift we observed between 15 µM and 1 mM Ca^2+^, where the E_rev_ in 1 mM Ca^2+^ is more negative. Also, In Figure 6D, a more depolarized holding potential should exacerbate the accumulation/depletion effect (if there is any), also pushing E_rev_ to more positive values. The data show the opposite.

We did not use a different ramp protocol because we predict the measured values will surely be different due to our hypothesis that membrane potential will drive ion movement into/away from the electric field. But for every experiment, we did switch solutions “back and forth” to make sure the results are reproduceable. Also, there is no correlation between E_rev_ with current magnitude (Figure 1—figure supplement 2 D, J). We will perform additional experiments to look into the effect of varying the ramp speed.

4) Subsection “Q559K channel ion selectivity varies with intracellular Ca^2+^ concentration”. The reversal potential of the QK/EQ mutant suggest this mutant is less Cl^-^ selective than the QK mutant alone. Thus, at an intermediate Po the selectivity is also intermediate. This is more consistent with a model where the pore can adopt multiple open states with different selectivity. What happens to the currents of the QK mutant alone (or WT) when measured at a Po~0.5? I do not think that the authors' conclusion that "This indicates either that TMEM16F ion selectivity is not correlated with its open state, or that the E667Q mutation circumvents the intermediate open state (if any) and causes the mutant channel to directly enter a fully-open conformation" is warranted by the data.

We thank the reviewer for making this point. . The "less negative shift" of Q559K_E667Q compared to Q559K can be attributed to the "cross talk" of the electrostatic field effect of Ca^2+^ in the binding pocket and ion entry into the permeation pathway, which has been shown in a recent paper for TMEM16A^10^. What we intended to indicate with this mutant is: Q559K channels in 15 µM Ca^2+^ or 1 mM Ca^2+^ are close to "full activation", and if the shift was due to the activation state, the "half-activation" mutant (i.e. Q559K_E667Q in 1 mM Ca^2+^)_should have more "positive" E_rev_ – which is not the case. We will take the advice by adding the data in 6 µM Ca^2+^ and revising the text to clarify the point.

5) Subsection “The switch of ion selectivity correlates with electrostatic change in permeating pathway”. If the shift in V_rev_ depends on divalent ions (Ca^2+^, Zn^2+^) then the shift should depend in a graded manner on their concentration. Did the authors test what happens when using intermediate increases between the 15 and 1000 μm concentrations shown?

We thank the reviewer for this question. Yes, Figure 3 shows our results in ~40 µM Ca^2+^. We will mention this point in the presentation of Figure 3.

6) Are divalent anions inert? Since the channel is permeable to anions under all conditions (even when it is in its "cation selective" state, its selectivity is modest), these ions can enter the pore and therefore should have the opposite effect of the divalent cations.

This is a very interesting question. However, TMEM16A is not permeable to divalent ions such as SO_4_^2-^, so we are not sure if this will work for TMEM16F Q559K. But nonetheless, we can try testing the reversal potential with SO_4_^2-^ in the solution.

7) The inactivation in Figure 4A is slow (tau~30s – 1 min). Why weren't the reversal potentials measured after reaching activation but before inactivation sets in? Is inactivation dependent on voltage? If the patches are held at a membrane voltage closer to reversal is there a similar effect?

We thank the reviewer for the insight. We did this experiment. Although the rundown of TMEM16F WT current magnitude in 1 mM Ca^2+^ is "slow", the change of its rectification is relatively fast (Figure 1—figure supplement 2 F, the trace in 45 mM NaCl was recorded ~5 s after solution change to 1 mM Ca^2+^). Combining these changes together, we cannot draw convincing conclusions from these data. But interestingly, we noticed that if the patch was exposed to 1 mM Ca^2+^ in 15 mM NaCl, the change of rectification is slower, enabling us to obtain data as in Figure 1_figure supplment 4. We speculate that the inactivation indeed depends on the difference between voltage and E_rev_, and more specifically, the amount of permeating cation. But this is beyond the scope of this manuscript.

8) I find the effects of Zn^2+^ confusing where they regard the rapid run-down that is PIP_2_ independent. Are there multiple run-down mechanisms?

We thank the reviewer for mentioning this point. From Figure 4—figure supplement 1, the Zn^2+^-induced rundown was reversed in ~1 minute. We thus believe Zn^2+^-induced rundown was not caused by degradation of PIP_2_, which is irreversible, although we are not sure whether Zn^2+^ can sequester PIP_2_ in a reversible manner. We will re-write the sentence to make it clear.

9) Discussion section. The authors state "However, TMEM16F does not have an "Ohmic" state: the channel always requires depolarization to be activated." What does this mean? That the pore is intrinsically rectifying? Ohmic state would refer to conductance, not gating.

We apologize for the confusing statement. TMEM16A in high Ca^2+^ concentration (>a few µM) stably binds to two Ca^2+^ ions so that its conductance is "Ohmic". For TMEM16F, the data indicate that it is always outwardly rectifying regardless of Ca^2+^ concentration, but we don't know whether the pore is "intrinsically rectifying" or Ca^2+^ ions are not stably bound without depolarization. We will rewrite the sentence to clarify this point.

Reviewer #2:[…] The authors results are very interesting, but I have several concerns.1) Experimental artifacts: To combat channel rundown and voltage-dependent activation, the authors have resorted to protocols with sizeable resting membrane currents and conductance, conditions that can easily lead to ion accumulation/depletion artifacts ("Calcium-calmodulin does not alter the anion permeability of the mouse TMEM16A calcium-activated chloride channel", JGP 2014 Jul; 144(1): 115-124.).I am sure the authors have thought carefully about this issue and probably already have data to exclude this possibility. For example, since the channel undergoes desensitization, they should have recordings in which early ramps have a much larger conductance than later ramps recorded under identical conditions (e.g. [Ca^2+^], holding voltage). If the reversal potential for each condition is essentially independent of the holding current/conductance, ion accumulation/depletion artifacts are unlikely. Reporting key experimental details (e.g. complete protocol and order in which conditions/ramps were performed, leak subtraction protocol, series resistance compensation, composition of 15mM NaCl solution, control experiments to exclude effects of Ca^2+^, HEPES, EDTA permation, etc.) would help address these concerns.

We thank the reviewer for these helpful comments and his trust that we have thought carefully about the issues raised. We will take the advice and explain our methods with more details in the revised manuscript. As we stated in the responses to all reviewers, the concern regarding the ion accumulation/depletion artifacts can be addressed for the following reasons:

For most recordings, we switched solutions "back and forth" to make sure traces in the same solution reversed at the same voltage despite the change of current magnitude. We also successively recorded multiple traces in the same solution to make sure there was no drift. Between different cells, there was no correlation between E_rev_ and current magnitude (Figure 1—figure supplement 2 D, J).

It is important to clarify that the main conclusion of our study is not affected by any deviation (even if there is actually a deviation) caused by the ion accumulation or depletion:

We used a hyperpolarizing ramp from the holing potential at +80 mV (Figure 1—figure supplement 2). This will cause Na^+^ efflux and/or Cl^-^ influx, potentially leading to Na^+^ accumulation on the extracellular side and Cl^-^ accumulation on the intracellular side (or intracellular Na^+^ depletion and extracellular Cl^-^ depletion). All of these possibilities should push E_Na_ and E_Cl_ to “more positive” values, leading to a positive shift of E_rev_ (as observed in the cited JGP paper^1^). This is opposite to the difference between 15 µM and 1 mM Ca^2+^, where the E_rev_ in 1 mM Ca^2+^ is more negative. Also, In Figure 6D, a more depolarized holding potential should exacerbate the accumulation/depletion effect (if there is any), also pushing E_rev_ to more positive values. The data show the opposite.

2) Proposed mechanisms for holding voltage changing ionic selectivity:The authors seem to be proposing that Ca^2+^ (and perhaps also Na^+^) ions driven into the permeation pathway by the electric fields/driving force during the holding/activation voltage somehow modify permeation during the subsequent voltage ramp step. I have two concerns with this idea. Firstly, the authors have previously reported that TMEM16F is quite permeable to calcium. Larger depolarizations will drive more calcium out through the channel which could lower the calcium concentration on the intra-cellular side of the permeation pathway, rather than increasing it. Secondly, for the permeation pathway to act as a channel, ions must be able to enter and leave it on the microsecond timescale and thus one would expect rapid turnover of all permeating ions during the voltage ramp. For the channel to "remember" the holding voltage, it seems like voltage would either have to induce a conformational change or modulate the binding of ions to specific sites that turnover slowly enough to remain present during the subsequent ramp.

We thank the reviewer for the question. First, although the Ca^2+^ permeation may potentially reduce Ca^2+^ concentration at the intracellular interface of the pore, it is not fast enough to deplete Ca^2+^. This can be inferred from our data in a previous publication^13^: holding the membrane patch at a more depolarized membrane potential increases Ca^2+^ sensitivity of TMEM16F. If local intracellular Ca^2+^ depletion took place at a faster rate than Ca^2+^ entry into the membrane electric field, greater depolarization would lead to an increase of the EC_50_ for Ca^2+^ activation, which is opposite to what we found in our previous study. We will make this point clearly in the revised manuscript.

The second question concerns the structural basis for TMEM16F ion selectivity shift. The model proposed by the reviewer is based on a "conventional" selective filter exemplified by that in potassium channels, in which a few permeating ions "line up" and move along the pore, though multiple ions may still reside in a wide vestibule below the selectivity filter, as evident from the ability of magnesium and polyamines to cause inward rectification of Kir channels. At this point we do not have a TMEM16F open-channel structure, but we might infer from the structure of TMEM16A. Instead of a conventional selective filter similar to that in voltage-gated ion channels, the TMEM16A permeation pathway has amino acids that are critical for ion selectivity distributed along the pore; with a few narrow constrictions, the permeation pathway has a wide vestibule^14-16^. This structural feature is compatible with the possibility that ions can enter the permeation pathway and "resident at specific sites" without "leaving the pore", but still allow other permeating ions to move "beside them", reminiscent of schemes included in early editions of Bertil Hille’s book.

Reviewer #3:

*[…] While resolving the controversy in the cation/anion selectivity of TMEM16F is important, the authors' conclusion that the controversy results from different recording conditions used in various labs (such as different holding voltages, ionic conditions, or Ca^2+^ concentrations used) is a little bit hand waiving. In the literature, the permeability difference in TMEM16F between Na^+^ and Cl^-^ is not huge. Even accepting the results in this study as face value, the TMEM16F's* P_Na_/P_Cl_*ratios range from 0.5 to 2 (Figure 5D). It is therefore strange that the authors would call such a several-fold difference of Na^+^/Cl^-^ permeability as Na^+^ selective (or Cl^-^ selective in the other way around). Even worse, the changes of the* P_Na_/P_Cl_*permeability ratios reported here may actually be artefacts.*

The reviewer believes that the discrepancy regarding P_Na_/P_Cl_ in previously published literature as well as the shift in our study "is not huge". In fact, the moderate permeability ratio is nonetheless physiologically important. In a hypothetical cell whose Cl^-^ equilibrium potential is -60 mV and extracellular / intracellular cations are Na^+^ / K^+^ respectively, a shift of P_Na_/P_Cl_ from 0.5 to 2 can cause the membrane potential shift from -22 mV to -9 mV (assuming that this channel is the predominant local membrane conductance, and its P_Na+_ = P_K+_), big enough to regulate some signaling molecules such as NMDA receptors, voltage-gated Ca^2+^ and Na^+^ channels. Given that TMEM16F is proposed to be on the dendritic spines of neurons^2^ and primary cilia of epithelial cells^3^, a *local* membrane potential change can be very consequential.

In the present study and the Yang et al. paper cited by the authors (Yang et al., 2012), there are at least two serious technical problems that could contribute to the controversy. In fact, the result from Yang et al., 2012 and the result in this paper, both from the same lab, already contradict each other. Here is a striking example: in the present study, the authors show that with a high concentration (1 mM) of Ca^2+^, when [NaCl]i is changed from 150 mM to 15 mM, the reversal potentials (in both Q559K and wild-type TMEM16F) shift from ~ 0 mV to ~ -15 mV (Figure 1). In Yang et al., 2012, a similar 10-fold change of [NaCl]i in the presence of 0.5 mM Ca^2+^ generated a right shift of the reversal potential to ~+20 mV (Figure 6D and 6G in Yang et al., 2012). Both experiments are excised inside-out patch recordings, and the solutions are similar (150 mM/15 mM vs 140 mM/14 mM). The [Ca^2+^]i in Yang et al. was 0.5 mM, which should be closer to 1 mM Ca^2+^ than to 15 µM Ca^2+^. Why did the authors obtain a left shift of the reversal potential while in Yang et al., 2012, the reversal potential shifted to the right?

We thank the reviewer for the scrutiny. Actually, this question has already been answered in this manuscript: membrane potential matters. In the Yang et al. paper^18^, "A voltage ramp from −80 to +100 mV with a rate (dV/dt) = 0.36 V/s was used to determine the reversal potentials for the wild-type mTMEM16A and mTMEM16F channels" – this is a slow depolarizing ramp. By contrast, in the current manuscript, "The traces were recorded with a hyperpolarizing ramp from +80 mV to -80 mV (-1 V/s) following holding at +80 mV" – a fast reverse ramp. The protocol used in our manuscript reflects the channel permeability when it is at a more depolarized membrane potential. Inspired by review 2, we will add one experiment to show that TMEM16F can briefly "memorize" the ion selectivity . Also, please note that in Figure 6, in hundreds of μM Ca^2+^, the “conditioning membrane potential” impacts significantly on the measurements of reversal potential.

Given that TMEM16F (and most of the mutants) is outwardly rectifying, in the experiments shown in this manuscript, we mostly performed reversal ramps to obtain "more-opened" current and minimize the contamination by endogenous current. Due to the different experiment protocols used in the two studies, it is understandable that the measurements are different.

Two technical issues in both the current study and Yang et al., 2012, are not seriously dealt with.The first serious technical problem is the junction potential correction. Throughout the whole paper of Yang et al., 2012, there was no description of junction potential correction. In the present study, the authors mention that "the liquid junction potentials for 15 mM NaCl, (and) 45 mM NaCl were -1.7 mV and -1.0 mV respectively.……". I suppose that the authors calculated these values with respect to the 150 mM NaCl solution. If so, these values are severely under-estimated. In fact, using the same calculator (Clamplex), the value of liquid junction potential for 15 mM NaCl (with respect to 150 mM NaCl) is -11 mV (I confirm this value by actually measuring the liquid junction potential between 15 mM NaCl and 150 mM NaCl solution). Since the data points in the figures (For example, Figure 1F and G) are not corrected for the junction potential, the reversal potential of ~ -15 mV in 1 mM Ca^2+^ (see Figure 1G) should only be ~ -4 mV (with a 10-fold NaCl gradient). The reversal potential shift in Q559K in 15 µM Ca^2+^ in the paper is ~ +10 mV (Figure 1G). After junction potential correction, this should be ~+20 mV.

The reviewer might have neglected that we formed membrane seal in 150 mM NaCl-based solutions and used a 3 M KCl agar bridge as a reference electrode (so-called "salt bridge”) in both papers. The reviewer might have obtained the result (-11 mV) with the following parameters:

**Author response image 2. respfig2:** 

With our experimental protocol including forming giga-ohm seal in 150 mM NaCl and using a salt bridge during solution exchange, the correct setting should be:

**Author response image 3. respfig3:** 

This significantly reduces the liquid junction potential. With this setup involving a salt bridge, it is acceptable for Yang et al. not to correct it^18^.

The second and even more problematic issue is a lack of background current subtraction. Throughout the whole manuscript (and also in Yang et al., 2012), I do not see a description of background current subtraction for the presented I-V curves (and thus the measured reversal potentials). In the absence of TMEM16F conductance, the HEK293 cells have a cation-selective endogenous conductance. The authors actually show this fact from the background current measurement where the reversal potentials under 150 mM/15 mM NaCl condition were ~+20 mV (in 1 mM Ca^2+^) and ~ +60 mV (in 15 µM Ca^2+^) (see B and C in Figure 1—figure supplement 1). The conductance of the untransfected cell patches includes the leak conductance and the conductance from the endogenous channels/transporters. If we sum the two as the background conductance (abbreviated as gb), and the conductance from TMEM16F (the channel conductance) is denoted as gc, the measured reversal potential (E_rev_) should be a weighted sum of the reversal potential of gc (abbreviated as Ec) and that of gb (abbreviated as Eb) according to the following equation:E_rev_= [gc/(gc+gb)] × Ec + [gb/(gc+gb)] × Eb

This equation, derived from the chord conductance equation, assumes that the conductance (gc and gb) is constant at all voltages, which is not always applicable here. Nonetheless, the double mutant, Q559K_G615A, with reduced outward rectification (Author response image 1), is close to this situation. Based on this equation, the measurement of the reversal potential in 15 µM Ca^2+^, which implicates higher Na^+^permeability, is not affected as much as the reversal potential in 1 mM Ca^2+^, which implicates higher Cl^-^ permeability, whose real value should be even more “negative” (since the background current is cation-selective, Figure 1—figure supplement 3B,C). This is shown in Author response image 4 which is adapted from Figure 4C. Thus, our conclusion that there is a shift in E_rev_ is upheld.

**Author response image 4. respfig4:** 

But nonetheless, we agree that it is necessary to reduce the contamination by gb, which we were very careful with. We also agree that currents with strong outward rectification are more susceptible to background contamination – we have explained how we performed "quality control" in the "response to all reviewers", and we will reinstate these points in the following response.

Reducing [NaCl] or Ca^2+^ will reduce the TMEM16F conductance (gc), and therefore, E_rev_ will approach Eb (which is +20 mV or +60 mV). When the gc is much larger than gb (this is the case when TMEM16F is activated by 1 mM Ca^2+^), the contribution of gb to the measured E_rev_ is less significant (as shown in Figure 1—figure supplement 1A, blue and grey curve). However, when gc is small relative to gb, the measured E_rev_ will be severely contaminated by Eb. The example is no better than the one provided by the authors in Figure 6, where when the holding voltage is more positive (the TMEM16F conductance is larger because of its outward rectification property), the reversal potential is less positive (less contaminated by Eb). This phenomenon is consistent with all the results throughout the paper. Various manipulations in the paper (including several mutations or even experiments in TMEM16A) would not change this trend-the larger the gc, the less positive the measured E_rev_.

We respectfully disagree with the statement that "Reducing [NaCl] or Ca^2+^ will reduce the TMEM16F conductance (gc)". Please note that Q559K current activated by 15 µM Ca^2+^ and by 1 mM Ca^2+^ are similar in magnitude (Figure 1B, D). Although exposure to >30 µM Ca^2+^ leads to current desensitization due to loss of PIP_2_, (Figure 1B, Also Ref^13^) this is not an issue since we performed all the experiments in 15 µM Ca^2+^ before patch exposure to high Ca^2+^ in experiments throughout the whole manuscript (We formed the excised patch in 0 Ca^2+^ and carefully prevented the patch from exposure to Ca^2+^ > 20 µM^13^). The reviewer might have misinterpreted Figure 1F: the current activated by 15 µM Ca^2+^ in 15 mM NaCl at +80 mV “looks” small, but that is because the driving force of cation efflux is smaller (Ohm’s law: current is proportional to the product of driving force and conductance), while there is no evidence showing that the conductance is small. In 1 mM Ca^2+^, the current at +80 mV is not reduced simply because now the outward current is mainly carried by Cl^-^ influx. This does not disprove our hypothesis, but instead, is consistent with our conclusion.

The reviewer uses Figure 6 (Figure 7 in the revised manuscript) as an example. Figure 6D (now Figure 7A) shows that holding at different conditioning voltages changes the reversal potential recorded with the hyperpolarizing ramp protocol. The traces were obtained from the same patch, so the contaminating gc is universal for the four traces. In whatever way the hypothetical background current is drawn, the shift of E_rev_ is legitimate (shown in Author response image 5).

**Author response image 5. respfig5:** 

Without subtracting the background current obtained in the absence of Ca^2+^, how large the total current should be so that the measured E_rev_ is good enough to reflect the true TMEM16F reversal potential (Ec)? The equation above indicates that gc needs to be larger than 90% of the total conductance (gc + gb) in order to have a measurement where <10% of Eb (namely +2 mV to +6 mV, according to the measured Eb) is added to the measured reversal potential.

We thank the reviewer for the friendly reminder. We have explained our detailed methods in the “responses to all reviewers” that we made sure gb is <10% as the reviewer points out. Nonetheless, performing background correction will expand the range of “E_rev_ shift”. This can be illustrated in the Author response images 4 and 5.

The authors show that the background current from untransfected cell patches at +80 mV is 50-80 pA (Figure 1—figure supplement 1B and C). If the total current at +80 mV is 500 pA (0.5 nA) in the solution with 15 µM Ca^2+^ and 15 mM NaCl, as in many of the authors' recording traces, the conductance of TMEM16F at +80 mV is ~ 9-10 fold that of the background current. However, since WT TMEM16F and Q559K mutant are outwardly rectifying, their conductances are quickly reduced by several fold near the reversal potential (the outward rectification is even more severe in low NaCl and low Ca^2+^ condition). Therefore, the 9-10 fold gc/gb ratio at +80 mV may reduce to 2-3 near Ec (gb is constant at different voltages). Suppose the true value of Ec is 0 mV in the 150 mM/15 mM NaCl condition (namely, P_Na_/P_Cl_ =1), and the value of Eb is +60 mV (as shown in the supplementary figure), a gc/gb ratio of 2-3 will give a measured reversal potential of +20 mV – +30 mV. If the total current at +80 mV is less than 0.5 nA, the situation will be even worse. Interestingly, the reversal potentials measured in 150 mM/15 mM NaCl condition in many experiments in this manuscript are in this range.

We thank the reviewers for pointing this out. We explained our detailed methods in responses to all reviewers, but we would like to summarize here:

· We included the recordings with Ca^2+^-activated current > 1000 pA, which practically were ~ 2000 pA (Figure 1—figure supplement 2), 20 fold the size of the background current (<100 pA) (Figure 1—figure supplement 2A, D, G, J). The "0.5 nA" mentioned by the reviewer might refer to the current in 15 mM NaCl (blue traces in Figure 1—figure supplement 2B, E, H, K), which might actually reflect a reduction of driving force for cation efflux, rather than a reduction of conductance.

· We did take rectification into consideration and this consideration did not change the conclusion. We made sure the tail current recorded at + 80 mV after completion of the reversal ramp is still ~5 fold of the background current (Figure 1—figure supplement 2C, I, L). Also for this reason, we did not analyze WT current activated by 1 mM Ca^2+^ (Figure 1—figure supplement 2F).

· In some of the recordings (such as Figure 1—figure supplement 2B), the current in 15 mM NaCl 15 µM Ca^2+^ at 0 mV is ~-200 pA, and it would not have been eliminated by background subtraction.

· Q559K_G615A circumvents the concern of outward rectification.

The above example therefore emphasizes the importance of subtracting the background current in order to evaluate EC more precisely. However, I am only using the data provided by the authors to make the above estimate. Since the background currents vary from patch to patch, the easy way to minimize the contribution of the background conductance is to subtract the current obtained in the absence of Ca^2+^ from current obtained in the presence of Ca^2+^. By doing background current subtraction, the authors would eliminate the shift in E_rev_ caused by the background conductance and thus isolate the shift in E_rev_ caused by Ca^2+^.

We thank the reviewer for this suggestion. We did think about using background subtraction but did not eventually perform it, because it will bring about new issues. The reviewer suggests we should look at the Ca^2+^-activated component. But this is based on the assumption that Ca^2+^ does not have effects on background current, which is NOT the case. Judging from Figure 1—figure supplement 3, we noticed that there is a cation selective current endogenous to HEK cells, and this current is smaller in 1 mM Ca^2+^, suggesting that Ca^2+^ "blocks" the background cation current. We have observed this result frequently when we recorded from untransfected cells (See Author response image 6).

**Author response image 6. respfig6:** 

Performing background subtraction will "over-subtract" a cation component (~40 pA) in 1 mM Ca^2+^, making the 1-mM-Ca^2+^-evoked current appear "more anion-selective". On the other hand, traces recorded from the same patch have the same background current, but the shift was legitimate (shown in Author response image 5). Thus, we chose to take a more cautious and conservative action: NOT to perform background subtraction. Our Figure 1_figure supplement 3F shows the extrapolation to illustrate that background subtraction sometimes can be problematic.

It is interesting that similar background current contamination of the reversal potential measurement is also shown in TMEM16A and its K584Q mutant in this manuscript (Figure 6—figure supplement 1). However, it appears that the authors have not noticed this problem. Taking K584Q recordings as an example (see Figure 6—figure supplement 1), the K584Q conductance is bigger in 1 mM Ca^2+^ than in 0.4 µM Ca^2+^. Therefore, the reversal potential is more shifted to the right in 0.4 µM Ca^2+^. Here, the measured reversal potentials (10 fold NaCl gradient) of K584Q in 0.4 µM and 1 mM Ca^2+^ are both far from the perfect Cl^-^ Nernst potential. It has already been shown (see Supplementary figure 2 in Jeng et al., 2016) that after background current subtraction, the K584Q mutant, like the wild-type TMEM16A, is perfectly Cl^-^ selective in 20 µM Ca^2+^. In other words, based on the measured reversal potentials, K584Q in 20 µM Ca^2+^ in Jeng et al., 2016, is more Cl^-^ selective than the K584Q current activated by 1 mM Ca^2+^ shown in this paper. The key difference between the two studies is the background current subtraction.

We thank the reviewer for this comment. The data regarding whether TMEM16A is strictly or moderately selective for Cl^-^ over Na^+^ also has discrepancy among different groups, and several groups report that TMEM16A is moderately selective for Cl^-^ over Na^+^, not perfectly meeting the Cl^-^ Nernst potential^11,19^. In our recording of WT TMEM16A activated by 1 µM Ca^2+^ or in 1 mM Ca^2+^, the reversal potential is only shifted by ~15 mV in 45 mM NaCl (P_Na_/P_Cl_ ~ 0.2, osmolarity balanced with mannitol), different from the value reported by Jeng et al. (by ~30 mV). TMEM16A in 1 mM Ca^2+^ is linear and 60 times bigger than endogenous current, so background current subtraction will not account for the discrepancy. Jeng et al. had a very insightful discussion in their JGP paper:

· We suspect that differences in experimental methods, for example, using NMDG-SO_4_ versus sucrose to replace intracellular NaCl (Jeng et al., Lim et al), and/or analyses (such as subtraction of the endogenous leak current; Figure S2) may explain the discrepancy.

Given the above example, it is more likely that the "key difference" is "using NMDG-SO_4_ versus sucrose to replace intracellular NaCl". Notice that it is consistent with the idea conveyed in this manuscript: the intracellular ionic strength matters. NMDG might be still able to enter the electric field even though it cannot go through the channel, and it could account for the high anion permeability as recorded by Jeng et al.

For K584Q, although the current is small, it is nonetheless >10 times bigger than the background current. Since Ca^2+^ inhibits the cation-selective current endogenous to HEK cells (Figure 1—figure supplement 1B, C and previous paragraphs), background subtraction might cause new issues: The minor cation permeation (<1/10) might have been interpreted as background (which is already inhibited by Ca^2+^), and thus the corrected P_Na_/P_Cl_ will be smaller than the actual value. "The key difference between the two studies" (our study and Jeng et al) is in fact more than "background current subtraction". These discrepancies can be attributed to:

· Ca^2+^ concentration. The lowest Ca^2+^ concentration in Jeng et al. is 20 µM, while we tested 300 nM Ca^2+^. Note that 400 nM intracellular Ca^2+^ is a critical point where most TMEM16A channels are bound to 1 Ca^2+^ ion^11^, while in 20 µM Ca^2+^, TMEM16A binds to 2 Ca^2+^ ions. The electrostatic field along the TMEM16A permeation pathway is significantly different between the one-Ca^2+^-bound state and two-Ca^2+^-bound state^10^. Thus, in the experiments by Jeng et al., the permeation pathway might have been strongly influenced by the positive charges from two Ca^2+^ ions (they used 20 µM), overriding the effect of the mutation.

· Experimental details. Our unpublished data suggest that TMEM16A K584Q quickly runs down in 1 mM Ca^2+^, a feature mirroring the slower rundown of TMEM16F Q559K. The mechanism and consequence of rundown have not been studied. Thus, before we figure these out, we were careful about the experimental conditions such as forming the excised patch in 0 Ca^2+^ toprevent PIP_2_ degradation catalyzed by exposure to >20 µM Ca^2+^. Our methodology might "protect" some phenotypes that require PIP_2_. We infer that Chen lab did not make sure that the measurement of E_rev_ was performed without any prior exposure of the patch to >20 µM Ca^2+^ (a condition where PIP_2_ starts to degrade), judging from a recent paper by Chen lab using 900 µM Ca^2+^ as a reference condition to measure TMEM16F gating and ion selectivity^9^. It is thus not surprising that they got results different from our results and from a recently deposited preprint^12^.

With these technical flaws, the reported reversal potentials in this manuscript are simply not correct. I am not sure if the authors have measured background I-V curve in every recorded patch. If not, I do not think the problem can be corrected within a short time. If these two technical problems are corrected, I anticipate that conclusion of the paper is probably not going to hold.

We have demonstrated that we had thought about the two technical issues raised by reviewer 3 as we performed experiments. We reduced the liquid junction potential by using a 3 M KCl salt bridge. We have provided experimental details and performed extrapolation to prove that our conclusion is valid regardless of background subtraction, and we also provide recordings of the background current in the presence or absence of Ca^2+^ to illustrate the point that background subtraction might in fact bring about new issues.

[Editors' note: below is the authors’ response to the decision letter following their appeal.]

The editors and reviewers have read and discussed your rebuttal. It is important to point out that our initial concerns remain, and we are not convinced that your plan for revision will satisfy those concerns. However, we are willing to reconsider a revised manuscript after you have had a chance to do the additional experiments.

Overall revisions:

We have performed extensive experiments and added the following figures to our manuscript:

Figure 2: TMEM16F wild type and Q559K ion permeability ratios measured in 6 µM and 15 µM Ca^2+^, with background (recorded in Ca^2+^-free 15 mM NaCl solution) subtraction.

Figure 3: TMEM16F Q559K ion permeability ratios during current rundown/desensitization in 36 µM Ca^2+^, with background (recorded in Ca^2+^-free 15 mM NaCl solution) subtraction.

Revised Figure 1—figure supplement 4: TMEM16F Q559W ion permeability ratios in 15 µM and 1 mM Ca^2+^, with background (recorded in Ca^2+^-free 15 mM NaCl solution) subtraction.

Revised Figure 5: TMEM16F wild type ion permeability ratios in 10 µM and 1 mM Gd^3+^.

Figure 6—figure supplement 1: TMEM16F wild type and Q559K recorded in 15 mM NaCl balanced with NMDG-MES or NMDG_2_-SO_4_.

Figure 6—figure supplement 2: TMEM16F wild type recorded with 15 mM NaCl in the pipette solution and 150 mM NaCl in the bath.

We have also added Figure 1—figure supplement 2, Figure 1—figure supplement 3, Figure 4—figure supplement 2 to illustrate the experimental details and how we performed "quality control". All the results support the notion that TMEM16F permeability to Cl^-^ increases when intracellular Ca^2+^ is elevated. We hope these additional experiments and explanations will clear the concerns regarding technical issues.

Inspired by the editors, reviewer 1 and reviewer 2, we have performed experiments (now shown in Figure 7 F~H, Figure 8—figure supplement 1), renamed our mechanism as a “dual-gating model” for the purpose of a comparison with that of TMEM16A. We did not draw a definite conclusion from these experiments, but we hope these results will trigger discussions and new ideas among readers after the paper is published.

We also took the editors’ suggestion to reduce the discussion of physiological implication and focus more on the characterization of channel biophysical properties in the text. In the revised manuscript, we briefly discussed the physiological implication of the ion selectivity shifting in the discussion. As was mentioned in our correspondence with editors regarding the case of BK channel, we believed that a clear understanding of the channel biophysical properties will prompt the elucidation of important physiological functions of the channel. Thus, we wish these results (including the main results of ion permeability shifting in response to Ca^2+^ and voltage, and the addition experiments about channel activation by Zn^2+^ and Gd^3+^, the function of TM6 in gating) would be available to researchers working on TMEM16 and other channels with structural similarities (such as TMC and OSCA).

The following are several points that you should consider in preparing the revised manuscript.1) We remain concerned that leak currents could contribute in a non-linear fashion to the observed shifts in V_rev_. The data provided in your rebuttal does not address this issue, as in different ionic concentrations the shape and V_rev_of the leak will change. We suggest that you bracket each IV measurement in any solution containing calcium with measurements of the background IV (i.e. 0µM Ca^2+^) in solutions with identical NaCl concentration so that the voltage dependence of the total and leak currents can be directly compared. We understand that there is no perfect way to identify and subtract leak currents for TMEM16F, but by showing raw and leak corrected data it will be possible to estimate the total and leak conductances at the reversal potential and thereby gauge the accuracy and reliability of the V_rev_calculations. Since this does not appear to have been done in your initial experiments, we would request that they show a set of experiments between the extremes of low and high current density where you do these experiments, perhaps using a couple of different constructs.

We thank the editors for this practical requirement. We adopted the suggested experimental procedures when we performed new experiments for WT and Q559K (in 6 µM, 15 µM, 36 µM and 1 mM Ca^2+^, Figure 2 and 3) and for the new construct, Q559W (Figure 1—figure supplement 4). We also applied this strategy to our experiments where 15 mM NaCl was balanced with salts (NMDG-MES and NMDG_2_-SO_4_) instead of mannitol (Figure 6—figure supplement 1). We would like to thank the editors for this suggestion, which enabled us to further firm up the conclusions drawn from Figure 3, in which for each patch all the currents recorded in 36 µM Ca^2+^ crossed with the background current at the same point, suggesting E_rev_ did not shift with rundown or rectification state.

*2) As you point out, the current in your experiments is typically quite high, in the range of 1-8 nA when activated with calcium at +80mV. Although increasing current density diminishes the contribution of leak currents, currents of this magnitude will cause substantial voltage errors due to series resistance and will also alter ion concentrations (via ion depletion and accumulation). The plots of the current magnitude vs V_rev_shift in Author Response Image 1D of your rebuttal only partially addresses this concern since the conductance of TMEM16F is nonlinear and you have not reported the TMEM16F and leak conductances at the reversal potential. Furthermore, the data* in Author Response Image 1D *shows that there is a ~2-fold variability in the V_rev_measurements (between ~15 and ~35 mV), in a similar range to the changes in V_rev_that lead you to conclude that the ion selectivity of TMEM16F changes. We are not certain how best to resolve this issue, but we suggest that it might be helpful to provide V_rev_measurements as outlined above for both low and high calcium and to extend the range shown in Figure 1D to somewhat lower current density, in the range of 500 pA. While these might be affected more by background leak currents, they might also serve to validate the conclusions by providing an empirical calibration of the role of the leaks and other errors.*

We thank the editors for the suggestion. The current magnitude of Q559W happened to be around ~500 pA, which we thus utilized to perform the measurement as requested. Regardless of background subtraction, the shift was legitimate (Figure 1—figure supplement 4). Please note that the current in 15 µM Ca^2+^ after background subtraction reversed at a more positive potential than current in 1 mM Ca^2+^ without background subtraction, confirming that the permeability ratio transition should be reliable. But there are variations across different samples even for the same batch of cells, recorded on the same day. Thus, "for each experiment in the following study, we performed paired-comparison within the same patch to avoid ambiguities arising from large variations across different recordings."

3) We suggest that you rephrase how you describe the ion selectivity of TMEM16F. Assuming a clear shift in V_rev_can be unambiguously demonstrated, it would be more appropriate to refer to the protein as a poor ion selective channel that slightly changes its ionic preference (rather than a sodium selective channel that becomes chloride selective).

We have rephrased the statements as suggested. We now used "change of permeability ratio" in most parts of data interpretation. We only use "ion selectivity" when we generally describe channel biophysics.

4) We think the potential physiological effects of the ion selectivity changes you report are overstated. You argue that the effects might induce a ~13 mV change in the resting potential of a cell where TMEM16F is the main conductance. However, we are unaware of conditions where a cell would be exposed to millimolar intracellular calcium and sustained large depolarizations. Moreover, the presence of other currents would significantly diminish the impact of small changes in the selectivity of TMEM16F, especially since in the -80 to +50 mV range (where the cell membrane potential is likely to be), TMEM16F has a very low Po (i.e. Figure 1—figure supplement 2D). Therefore, the physiological effects on a cell's membrane voltage -if any- will be considerably smaller than ~13 mV. We request that the physiological relevance should be removed from the Abstract and significantly toned down in the Discussion. The focus of the manuscript should be on the biophysical mechanism underlying the small shift in selectivity.

We thank the editors for the insight and have revised our manuscript as requested. Now the main text is focused on the biophysical mechanism, while the physiological implication was only briefly discussed at the end.

5) We suggest that you add a discussion about the origin of the discrepancy of the ion selectivity of TMEM16A and TMEM16F.

We thank the editors for the suggestion. We have added the discussion of the comparison between TMEM16A and TMEM16F, including but not limited to the aspect of ion selectivity.

Reviewer #1:[…] 1) TMEM16F is a scramblase, as such its main function is lipid transport to externalize PS to initiate blood coagulation and recognition of apoptotic cells by macrophages. The physiological relevance of TMEM16F-mediated ionic currents is -at the moment- unclear. Further, activation of TMEM16F currents require sustained depolarizations and high Ca^2+^ concentrations, while lipid scrambling requires only elevation of Ca^2+^, but not depolarization. Thus, it is not clear under what physiological conditions the self-braking mechanism enabled by the dynamic switch in selectivity might play a role. For example, does the membrane potential of platelets and/or apoptotic cells reach and maintain the depolarizations necessary for channel activation? Furthermore, the magnitude of the observed switch is relatively modest; in all conditions the channel is poorly selective between anions and cations, displaying only modest preferences for one over the other. In no cases do the currents approach ideal anion vs cation selectivity. Given the overall poor selectivity, I wonder how much of an effect would be caused by the observed switch in selectivity. Thus, while the phenomenon described here is biophysically interesting, I am not convinced of its physiological relevance. Some quantitation of the effects observed here within the context of a cell would be needed to establish a clear link to physiology.

We thank the reviewer for raising the question regarding the physiological relevance of our work. It is true that previous studies regarding TMEM16F functions are mostly focused on its function in phospholipid scrambling and vesiculation. But one possible reason that hampers the study of its ion channel functions is the ambiguity of its ion selectivity; the conflicting results might have prevented us from understanding the channel functions thoroughly. Thus, our characterization of its biophysical properties might help unravel the physiological significance involving channel function of TMEM16F. We have included a brief discussion in this regard in the manuscript.

2) It is not clear through what type of pore ions move in TMEM16F. It has been proposed that the TMEM16 scramblases form a lipid-lined pore through which ions can diffuse together with the lipids that are being scrambled (Whitlock and Hartzell, 2016; Jiang et al., 2017). If this were the case, then the selectivity of the channel would be affected by multiple factors such as lipid headgroups, voltage and mono- and di-valent ions. On the other hand, the TMEM16A homologue forms a more conventional, protein-delimited, pore. In this case, the observed changes in selectivity could reflect either a charge-screening effect (as the authors propose) or the ability of the channel's pore to adopt slightly different conformations. Further, most of the measurements are carried out using a mutant, Q559K, that affects both selectivity and gating of TMEM16F. I understand that the mutant was used because of the removal of run-down. It is unclear whether this construct can serve as a good model for the WT pore. The mutation could case subtle rearrangements in the pore rendering it more sensitive to environmental changes. While the present experiments provide evidence suggesting that the selectivity of TMEM16F changes, they shed little light into the mechanisms underlying such a switch.

We thank the reviewer for the question concerning the structural basis of TMEM16F and the representativeness of the Q559K mutant.

In the revised manuscript, we added a few experiments to show that the shift of permeability ratio also happens to wild-type TMEM16F, including: (1) Using Gd^3+^ to activate TMEM16F for reversal potential measurements. Gd^3+^, a trivalent ion, increases Cl^-^ accessibility to the permeation pathway likely due to its stronger influence of the electrostatic field, so that wild-type TMEM16F in 1 mM Gd^3+^ maintains conductivity to Cl^-^ at low membrane potentials. (2) Alternating the pipette solution (equivalent to extracellular solution) and the bath (intracellular solution) so that in 1 mM Ca^2+^ the current reversed at a positive membrane potential. In addition, we confirmed that the critical phenotypes as found in Q559K mutant channel could all be recapitulated in WT, including the shift of E_rev_ in accordance to ionic strength (Figure 6A), to voltage (Figure7D), and "transiently" to Ca^2+^ level (Figure 1—figure supplement 4). A recently published paper has also proposed using Q559 mutants to study the ionic transportation mechanism of TMEM16F^9^.

To consider whether the lipid headgroups affect the ion selectivity, we performed an additional experiment as in Figure3, showing that PIP_2_ does not play a significant role. TMEM16F current activated by Gd^3+^ - an ion that has not been reported to trigger scrambling – also shows a shifting permeability ratio. In addition, cells heterologously expressing Q559K proteins do not display scrambling activities^8^, although purified Q559K shows scrambling activities in liposomes^2^. Taken together, we believe the shifting of permeability ratio is independent of scrambling. However, we could not exclude the possibility that there could be "slightly different conformations" on top of the “charge screening effect”, so we rephrased our statement as "although it remains possible that TMEM16F might undergo miniscule conformational change at different Ca^2+^ concentrations that contributes to the selectivity transition". In fact, our discussion regarding the relationship between ion selectivity and scrambling function also includes this point.

3) A technical concern I have is that the measured currents are relatively large (~1 nA) and the stimulation protocols used are long. Can the authors rule out that the effects they see are due to ion accumulation/depletion effects? The currents in 15 μm Ca^2+^ are naturally smaller than those in 1 mM Ca^2+^, and therefore these effects are less pronounced leading to the observed differences. Did the authors use different directs of the ramp and/or tail current protocols to measure reversal potentials? Is there an effect on the direction of the ion switch (i.e. going from 150 NaCl to 15 NaCl is the same as going from 15 NaCl to 150 NaCl)? Do they observe a dependence of the magnitude of the effects as a function of the current in the patch?

We thank the reviewer for the helpful suggestion. The effects we observe are unlikely due to ion accumulation/depletion. We used a hyperpolarizing ramp following holing the membrane potential at +80 mV (See Figure 1—figure supplement 2). This will cause Na^+^ efflux and/or Cl^-^ influx, supposedly leading to Na^+^ accumulation on the extracellular side of the channel and Cl^-^ accumulation on the intracellular side (or intracellular Na^+^ depletion and extracellular Cl^-^ depletion). All of these should push E_Na_ and E_Cl_ to “more positive” values, leading to a positive shift of E_rev_. This is opposite to the shift we observed between 15 µM and 1 mM Ca^2+,^where the E_rev_ in 1 mM Ca^2+^ is more negative. Also, In Figure 6D, a more depolarized holding potential should exacerbate the accumulation/depletion effect (if there is any), also pushing E_rev_ to more positive values. The data show the opposite.

We thank the reviewer for the helpful suggestion. We have answered this question in our responses to all reviewers: the effects we observe are unlikely due to ion accumulation/depletion, which actually can result in an opposite shift from what has been observed in our study.

We did not use a different protocol because we predict the measured values will surely be different due to our hypothesis that membrane potential will drive ion movement into/away from the electric field. There is no correlation between E_rev_ with current magnitude (Figure 1—figure supplement 2D, J).

4) Subsection “Q559K channel ion selectivity varies with intracellular Ca^2+^ concentration”. The reversal potential of the QK/EQ mutant suggest this mutant is less Cl^-^ selective than the QK mutant alone. Thus, at an intermediate Po the selectivity is also intermediate. This is more consistent with a model where the pore can adopt multiple open states with different selectivity. What happens to the currents of the QK mutant alone (or WT) when measured at a Po~0.5? I do not think that the authors' conclusion that "This indicates either that TMEM16F ion selectivity is not correlated with its open state, or that the E667Q mutation circumvents the intermediate open state (if any) and causes the mutant channel to directly enter a fully-open conformation" is warranted by the data.

We thank the reviewer for making this point. We measured the E_rev_ values of WT TMEM16F and Q559K in 6 µM Ca^2+^ (a condition where Po is smaller than 0.5) and they were more positive than those in 15 µM Ca^2+^ respectively (revised Figure 2). What we intended to express is: If the transition to being more permeable to Cl^-^ (versus Na^+^) reflects different activation states, the "half-activation" mutant (Q559K_E667Q) should have more "positive" E_rev_ (than Q559K in 15 µM Ca^2+^, the latter in a "more fully" open state) - which is not the case. We have revised the text to clarify the point. The "less negative shift" of Q559K_E667Q compared to Q559K can be attributed to the "cross talk" of the electrostatic field resulting from Ca^2+^-binding to the binding pocket and ion entry into the permeation pathway, to which a similar result has been shown in a recent paper for TMEM16A.^10^

5) Subsection “The switch of ion selectivity correlates with electrostatic change in permeating pathway”. If the shift in V_rev_ depends on divalent ions (Ca^2+^, Zn^2+^) then the shift should depend in a graded manner on their concentration. Did the authors test what happens when using intermediate increases between the 15 and 1000 μm concentrations shown?

We thank the reviewer for this question. We revised Figure 2 and added a new Figure 3 to show there is a "dose-dependent" shift.

6) Are divalent anions inert? Since the channel is permeable to anions under all conditions (even when it is in its "cation selective" state, its selectivity is modest), these ions can enter the pore and therefore should have the opposite effect of the divalent cations.

This is a very interesting question. As part of the new experiments included in the revised manuscript, we compared NMDG-MES and NMDG-SO_4_, which did not show different impact, likely reflecting their inaccessibility to the permeation pathway.

7) The inactivation in Figure 4A is slow (tau~30s – 1 min). Why weren't the reversal potentials measured after reaching activation but before inactivation sets in? Is inactivation dependent on voltage? If the patches are held at a membrane voltage closer to reversal is there a similar effect?

We thank the reviewer for the insight. Although the rundown of TMEM16F WT current magnitude in 1 mM Ca^2+^ is "slow", the change of its rectification is relatively fast (Figure 1—figure supplement 2, the trace in 45 mM NaCl was recorded ~5 s after solution change to 1 mM Ca^2+^). Combining these changes together, we cannot draw convincing conclusions from these data. But interestingly, we noticed that if the patch was exposed to 1 mM Ca^2+^ in 15 mM NaCl, the change of rectification was slower, enabling us to obtain data as shown in Figure 1—figure supplement 4. We speculate that the inactivation indeed depends on the difference between voltage and E_rev_, and more specifically, the amount of permeating cation.

8) I find the effects of Zn^2+^ confusing where they regard the rapid run-down that is PIP_2_ independent. Are there multiple run-down mechanisms?

We thank the reviewer for mentioning this point. From Figure 5—figure supplement 1, the Zn^2+^-induced rundown was reversed in ~1 minute. We thus believe Zn^2+^-induced rundown was not caused by degradation of PIP_2_, which is irreversible, although we are not sure whether Zn^2+^ can shield the charges of PIP_2_ headgroups in a reversible manner. We have added this statement in the manuscript.

9) Discussion section. The authors state "However, TMEM16F does not have an "Ohmic" state: the channel always requires depolarization to be activated." What does this mean? That the pore is intrinsically rectifying? Ohmic state would refer to conductance, not gating.

We apologize for the confusing statement. We intended to express that TMEM16A conductance is Ohmic at high Ca^2+^ while TMEM16F is outward rectifying at all tested Ca^2+^ concentrations, which could be owing to the "intrinsically rectifying" (i.e. non-Ohmic) conductance and/or the voltage-dependent gating. The former is shown with an "instantaneous rectification index" ^12^, and our steady-state activation curve (Figure 1—figure supplement 1) shows the latter. We have re-written this part in the discussion.

Reviewer #2:[…] 1) Experimental artifacts: To combat channel rundown and voltage-dependent activation, the authors have resorted to protocols with sizeable resting membrane currents and conductance, conditions that can easily lead to ion accumulation/depletion artifacts ("Calcium-calmodulin does not alter the anion permeability of the mouse TMEM16A calcium-activated chloride channel", JGP 2014 Jul; 144(1): 115-124.).I am sure the authors have thought carefully about this issue and probably already have data to exclude this possibility. For example, since the channel undergoes desensitization, they should have recordings in which early ramps have a much larger conductance than later ramps recorded under identical conditions (e.g. [Ca^2+^], holding voltage). If the reversal potential for each condition is essentially independent of the holding current/conductance, ion accumulation/depletion artifacts are unlikely. Reporting key experimental details (e.g. complete protocol and order in which conditions/ramps were performed, leak subtraction protocol, series resistance compensation, composition of 15mM NaCl solution, control experiments to exclude effects of Ca^2+^, HEPES, EDTA permation, etc.) would help address these concerns.

We thank the reviewer for these helpful comments and his trust that we have thought carefully about the issues raised. We have taken the advice and explained our methods with more details in the revised manuscript. As we stated in our responses to all reviewers, the concern regarding the ion accumulation/depletion artifacts can be alleviated by the experimental protocols and controls implemented in this study so our conclusion is valid.

2) Proposed mechanisms for holding voltage changing ionic selectivity:The authors seem to be proposing that Ca^2+^ (and perhaps also Na^+^) ions driven into the permeation pathway by the electric fields/driving force during the holding/activation voltage somehow modify permeation during the subsequent voltage ramp step. I have two concerns with this idea. Firstly, the authors have previously reported that TMEM16F is quite permeable to calcium. Larger depolarizations will drive more calcium out through the channel which could lower the calcium concentration on the intra-cellular side of the permeation pathway, rather than increasing it. Secondly, for the permeation pathway to act as a channel, ions must be able to enter and leave it on the microsecond timescale and thus one would expect rapid turnover of all permeating ions during the voltage ramp. For the channel to "remember" the holding voltage, it seems like voltage would either have to induce a conformational change or modulate the binding of ions to specific sites that turnover slowly enough to remain present during the subsequent ramp.

We thank the reviewer for the question. First, although the Ca^2+^ permeation may potentially reduce Ca^2+^ concentration at the intracellular interface of the pore, it is not fast enough to deplete Ca^2+^. This can be inferred from our data in a previous publication^13^: holding the excised patch at a more depolarized membrane potential increases Ca^2+^ sensitivity of TMEM16F. If local intracellular Ca^2+^ depletion took place at a faster rate than Ca^2+^ entry into the membrane electric field, greater depolarization would lead to an increase of the EC_50_ for Ca^2+^ activation, which is opposite to what we found in our previous study. We have added this sentence into our manuscript: "The reduction of EC_50_ of Ca^2+^ activation by depolarization is also observed in TMEM16F, although TMEM16F permeates Ca^2+^, suggesting Ca^2+^ efflux is not fast enough to deplete Ca^2+^ in the electric field at the intracellular side."

The second question concerns the structure basis for TMEM16F ion selectivity shift. The model proposed by the reviewer is based on a "conventional" channel selectivity filter exemplified by that in potassium channels, in which a few permeating ions "line up" and move along the pore. At this point we do not have a TMEM16F open-channel structure, but we might infer from the structure of TMEM16A and the possibly closed TMEM16F. Instead of a conventional selectivity filter similar to that in voltage-gated ion channels, the TMEM16 permeation pathway has amino acids that are critical for ion selectivity distributed along the pore; with a few relatively narrow constrictions, the permeation pathway has a wide vestibule^10,14-16^. This structural feature is compatible with the possibility that ions can enter the permeation pathway and remain as "resident at specific sites" without "leaving the pore", but still allow other permeating ions to move "beside them", reminiscent of schemes included in early editions of Bertil Hille’s book. We have considered this mechanism as an "electrostatic gating" mechanism instead of calling it a "selectivity filter", and we have added extensive discussion in the manuscript.

Reviewer #3:

*[…] While resolving the controversy in the cation/anion selectivity of TMEM16F is important, the authors' conclusion that the controversy results from different recording conditions used in various labs (such as different holding voltages, ionic conditions, or Ca^2+^ concentrations used) is a little bit hand waiving. In the literature, the permeability difference in TMEM16F between Na^+^ and Cl^-^ is not huge. Even accepting the results in this study as face value, the TMEM16F's* P_Na_/P_Cl_*ratios range from 0.5 to 2 (Figure 5D). It is therefore strange that the authors would call such a several-fold difference of Na^+^/Cl^-^ permeability as Na^+^ selective (or Cl^-^ selective in the other way around). Even worse, the changes of the* P_Na_/P_Cl_*permeability ratios reported here may actually be artefacts.*

We have taken the editors’ suggestion to refer to the channel as being “more Na^+^(or Cl^-^) permeable” instead of “selective”, and to reduce the discussion of physiological implication. Since the expression and localization of the channel is not known, we could not assert that such a change in channel permeability is “not huge”. In a hypothetical cell with Cl^-^ equilibrium potential of -60 mV and cation “equilibrium potential” (under the assumption that P_K_ = P_Na_) of 0 mV, a shift of P_Na_/P_Cl_ from 0.5 to 2 can alter the membrane potential from -22 mV to -9 mV (assuming that this channel is the predominant local membrane conductance), big enough to regulate some signaling molecules. It has also been reported that TMEM16F is localized in primary cilia, a special cellular compartment with elevated Ca^2+^ level and greater depolarization, providing a condition for TMEM16F to function dynamically. We believe our research can help unravel more physiological functions of TMEM16F.

In the present study and the Yang et al. paper cited by the authors (Yang et al., 2012), there are at least two serious technical problems that could contribute to the controversy. In fact, the result from Yang et al., 2012 and the result in this paper, both from the same lab, already contradict each other. Here is a striking example: in the present study, the authors show that with a high concentration (1 mM) of Ca^2+^, when [NaCl]i is changed from 150 mM to 15 mM, the reversal potentials (in both Q559K and wild-type TMEM16F) shift from ~ 0 mV to ~ -15 mV (Figure 1). In Yang et al., 2012, a similar 10-fold change of [NaCl]i in the presence of 0.5 mM Ca^2+^ generated a right shift of the reversal potential to ~+20 mV (Figure 6D and 6G in Yang et al., 2012). Both experiments are excised inside-out patch recordings, and the solutions are similar (150 mM/15 mM vs 140 mM/14 mM). The [Ca^2+^]i in Yang et al. was 0.5 mM, which should be closer to 1 mM Ca^2+^ than to 15 µM Ca^2+^. Why did the authors obtain a left shift of the reversal potential while in Yang et al., 2012, the reversal potential shifted to the right?

We thank the reviewer for the scrutiny. Actually, this question has already been answered in this manuscript: membrane potential matters. In the Yang et al paper, "A voltage ramp from −80 to +100 mV with a rate (dV/dt) = 0.36 V/s was used to determine the reversal potentials for the wild-type mTMEM16A and mTMEM16F channels" - this is a slow depolarizing ramp. By contrast, in the current manuscript, "The traces were recorded with a hyperpolarizing ramp from +80 mV to -80 mV (-1 V/s) following holding at +80 mV" – a fast hyperpolarizing ramp. The protocol used in our manuscript reflects the channel permeability when it is at a more depolarized membrane potential. Inspired by reviewer 2, we now show in the revised Figure 7 that TMEM16F can briefly "memorize" the ion selectivity at the preceding holding potential.

Given that TMEM16F (and most of the mutants) is outwardly rectifying, in the experiments shown in this manuscript, we mostly performed hyperpolarizing ramps to obtain "more-opened" current to minimize the contamination by endogenous current. Due to the different experimental protocols used in the two studies, it is understandable that the measurements yielded different results.

Two technical issues in both the current study and Yang et al., 2012, are not seriously dealt with.The first serious technical problem is the junction potential correction. Throughout the whole paper of Yang et al., 2012, there was no description of junction potential correction. In the present study, the authors mention that "the liquid junction potentials for 15 mM NaCl, (and) 45 mM NaCl were -1.7 mV and -1.0 mV respectively.……". I suppose that the authors calculated these values with respect to the 150 mM NaCl solution. If so, these values are severely under-estimated. In fact, using the same calculator (Clamplex), the value of liquid junction potential for 15 mM NaCl (with respect to 150 mM NaCl) is -11 mV (I confirm this value by actually measuring the liquid junction potential between 15 mM NaCl and 150 mM NaCl solution). Since the data points in the figures (For example, Figure 1F and G) are not corrected for the junction potential, the reversal potential of ~ -15 mV in 1 mM Ca^2+^ (see Figure 1G) should only be ~ -4 mV (with a 10-fold NaCl gradient). The reversal potential shift in Q559K in 15 µM Ca^2+^ in the paper is ~ +10 mV (Figure 1G). After junction potential correction, this should be ~+20 mV.

The reviewer might have neglected the fact that we formed the membrane seal with the patch electrode in 150 mM NaCl-based bath solutions and used a 3 M KCl agar bridge as a reference electrode (so-called "salt bridge”) in both papers. The reviewer might have obtained the liquid junction potential (-11 mV) with the following parameters:

**Author response image 7. respfig7:** 

The -11 mV actually reflects the offset of the commanding voltage for the liquid junction potential before sealing (rather than the liquid junction potential during solution exchange).

With our methods of forming giga-ohm seal in 150 mM NaCl and using a salt bridge at solution exchange, the correct setting should be:

**Author response image 8. respfig8:** 

This significantly reduces the liquid junction potential.

We have also included the experiments where the osmolarity of 15 mM NaCl solution was balanced with NMDG-MES and NMDG_2_-SO_4_ (Figure 6—figure supplement 1), where the liquid junction potentials are both smaller than 2 mV. Despite the potential complication resulting from NMDG permeation, the channel still displayed a shift in permeability ratio.

Last but not least, we respectfully point out that we do not agree with reviewer’s correction by “adding 10 mV to the measured value”, which coincidently is the method used in a recent paper^9^. Assuming that the authors of that paper did not use the salt bridge and that during the solution exchange there was an interface between the new and old baths, the “10 mV” should be subtracted from rather than added to the measured value (the situation comparable to the usage of a 150 mM NaCl reference electrode, picture below). If they had performed the correction correctly, they would have obtained the result showing that TMEM16F (recorded in 1 mM Ca^2+^) had reversed at a negative potential, corresponding to a channel more permeable to Cl^-^.

**Author response image 9. respfig9:** 

The second and even more problematic issue is a lack of background current subtraction. Throughout the whole manuscript (and also in Yang et al., 2012), I do not see a description of background current subtraction for the presented I-V curves (and thus the measured reversal potentials). In the absence of TMEM16F conductance, the HEK293 cells have a cation-selective endogenous conductance. The authors actually show this fact from the background current measurement where the reversal potentials under 150 mM/15 mM NaCl condition were ~+20 mV (in 1 mM Ca^2+^) and ~ +60 mV (in 15 µM Ca^2+^) (see B and C in Figure 1—figure supplement 1). The conductance of the untransfected cell patches includes the leak conductance and the conductance from the endogenous channels/transporters. If we sum the two as the background conductance (abbreviated as gb), and the conductance from TMEM16F (the channel conductance) is denoted as gc, the measured reversal potential (E_rev_) should be a weighted sum of the reversal potential of gc (abbreviated as Ec) and that of gb (abbreviated as Eb) according to the following equation:E_rev_= [gc/(gc+gb)] × Ec + [gb/(gc+gb)] × Eb

We thank the reviewer for expressing the concern of background contamination with considerable length (starting from here). We have included the new data together with the background current (the current recorded in 0 Ca^2+^, 15 mM NaCl, see responses to all reviewers) recorded from the same patch in the revised manuscript. The shift of E_rev_ could be observed regardless of background subtraction. We also have revised our manuscript with more detailed description about our methods of data acquisition and quality control. We believe our conclusion is now convincing given the additional controls and explanations. In the following parts, we will also list the supplementary points that we have addressed.

Reducing [NaCl] or Ca^2+^ will reduce the TMEM16F conductance (gc), and therefore, E_rev_ will approach Eb (which is +20 mV or +60 mV). When the gc is much larger than gb (this is the case when TMEM16F is activated by 1 mM Ca^2+^), the contribution of gb to the measured E_rev_ is less significant (as shown in Figure 1—figure supplement 1A, blue and grey curve). However, when gc is small relative to gb, the measured E_rev_ will be severely contaminated by Eb. The example is no better than the one provided by the authors in Figure 6, where when the holding voltage is more positive (the TMEM16F conductance is larger because of its outward rectification property), the reversal potential is less positive (less contaminated by Eb). This phenomenon is consistent with all the results throughout the paper. Various manipulations in the paper (including several mutations or even experiments in TMEM16A) would not change this trend-the larger the gc, the less positive the measured E_rev_.

We respectfully disagree with the statement that "Reducing [NaCl] or Ca^2+^ will reduce the TMEM16F conductance (gc)". Please note that the Q559K currents activated by 15 µM Ca^2+^ or by 1 mM Ca^2+^ are similar in magnitude (Figure 1B, D). The reviewer might have misinterpreted Figure 1F: the current activated by 15 µM in 15 mM NaCl at +80 mV “looks” small, but that is because the driving force of cation efflux is smaller, while there is no evidence showing that the conductance is small. In 1 mM Ca^2+^, the current at +80 mV is not reduced because now the outward current is mainly carried by Cl^-^ influx. This does not disprove our hypothesis, but instead, is consistent with our conclusion.

Without subtracting the background current obtained in the absence of Ca^2+^, how large the total current should be so that the measured E_rev_ is good enough to reflect the true TMEM16F reversal potential (Ec)? The equation above indicates that gc needs to be larger than 90% of the total conductance (gc + gb) in order to have a measurement where <10% of Eb (namely +2 mV to +6 mV, according to the measured Eb) is added to the measured reversal potential.

The reviewer’s equation, derived from the chord conductance equation, should only be applied if the conductance is constant at all voltages. But nonetheless, we made sure that the data such as those in Figure1, for which we did not perform background correction, met the criteria that gc is 9 times larger than gb (Figure 1—figure supplement 2). We would also like to remind the reviewer that according to this equation, given that gb is a cation current, we can extrapolate that for a linear current (such as Q559K_G615A, Figure 4, or wild type activated in 1 mM Gd^3+^,Figure 5), performing background subtraction will expand the shifting range of the E_rev_ when Ca^2+^ level changes.

The authors show that the background current from untransfected cell patches at +80 mV is 50-80 pA (Figure 1—figure supplement 1B and C). If the total current at +80 mV is 500 pA (0.5 nA) in the solution with 15 µM Ca^2+^ and 15 mM NaCl, as in many of the authors' recording traces, the conductance of TMEM16F at +80 mV is ~ 9-10 fold that of the background current. However, since WT TMEM16F and Q559K mutant are outwardly rectifying, their conductances are quickly reduced by several fold near the reversal potential (the outward rectification is even more severe in low NaCl and low Ca^2+^ condition). Therefore, the 9-10 fold gc/gb ratio at +80 mV may reduce to 2-3 near Ec (gb is constant at different voltages). Suppose the true value of Ec is 0 mV in the 150 mM/15 mM NaCl condition (namely, P_Na_/P_Cl_ =1), and the value of Eb is +60 mV (as shown in the supplementary figure), a gc/gb ratio of 2-3 will give a measured reversal potential of +20 mV – +30 mV. If the total current at +80 mV is less than 0.5 nA, the situation will be even worse. Interestingly, the reversal potentials measured in 150 mM/15 mM NaCl condition in many experiments in this manuscript are in this range.

Given that TMEM16F current is outwardly rectifying, the background contamination would cause a stronger deviation at a lower membrane potential. Thus, TMEM16F recorded in 1 mM Ca^2+^ would have reversed at an even more negative potential if the background current could be adequately removed, thus expanding the shifting magnitude (extrapolation, see Figure 1—figure supplement 3F). That was why we chose to “take a safer step by not performing background subtraction”. But nonetheless, for the new data included in the revised manuscript, we have demonstrated that our conclusion is upheld even with background subtraction.

The above example therefore emphasizes the importance of subtracting the background current in order to evaluate EC more precisely. However, I am only using the data provided by the authors to make the above estimate. Since the background currents vary from patch to patch, the easy way to minimize the contribution of the background conductance is to subtract the current obtained in the absence of Ca^2+^ from current obtained in the presence of Ca^2+^. By doing background current subtraction, the authors would eliminate the shift in E_rev_ caused by the background conductance and thus isolate the shift in E_rev_ caused by Ca^2+^.

We thank the reviewer for the suggestion. We have added the requested experiments with background subtraction into our revised manuscript. We also have discussed the disadvantage of background subtraction, namely expanding the shifting range in comparisons involving recordings in 1 mM Ca^2+^.

It is interesting that similar background current contamination of the reversal potential measurement is also shown in TMEM16A and its K584Q mutant in this manuscript (Figure 6—figure supplement 1). However, it appears that the authors have not noticed this problem. Taking K584Q recordings as an example (see Figure 6—figure supplement 1), the K584Q conductance is bigger in 1 mM Ca^2+^ than in 0.4 µM Ca^2+^. Therefore, the reversal potential is more shifted to the right in 0.4 µM Ca^2+^. Here, the measured reversal potentials (10 fold NaCl gradient) of K584Q in 0.4 µM and 1 mM Ca^2+^ are both far from the perfect Cl^-^ Nernst potential. It has already been shown (see Supplementary figure 2 in Jeng et al., 2016) that after background current subtraction, the K584Q mutant, like the wild-type TMEM16A, is perfectly Cl^-^ selective in 20 µM Ca^2+^. In other words, based on the measured reversal potentials, K584Q in 20 µM Ca^2+^ in Jeng et al., 2016, is more Cl^-^ selective than the K584Q current activated by 1 mM Ca^2+^ shown in this paper. The key difference between the two studies is the background current subtraction.

We thank the reviewer for the security. We have performed extensive experiments (Figure 8—figure supplement 1) to show that E_rev_ of TMEM16A recorded in low NaCl balanced with salts (NMDG-MES) is more negative than that in low NaCl with mannitol, with the former corresponding to a channel more selective for Cl^-^ (versus Na^+^). This was also reported in a paper in the same issue as the one cited by the reviewer, where the authors reported that TMEM16A recorded in low NaCl balanced with NMDG_2_-SO_4_ is more selective for Cl^-^ than that in low NaCl balanced with sucrose^16^. In other words, “the key difference” is the recording condition.

With these technical flaws, the reported reversal potentials in this manuscript are simply not correct. I am not sure if the authors have measured background I-V curve in every recorded patch. If not, I do not think the problem can be corrected within a short time. If these two technical problems are corrected, I anticipate that conclusion of the paper is probably not going to hold.

We have articulated why we initially did not perform background subtraction. We also included in the revised manuscript experiments with background subtraction, in which we demonstrated that the shift of E_rev_ still holds. We would like to take the opportunity to friendly express our thoughts to reviewer 3: In scientific research, it happens that different groups might come out with distinct results, under which circumstances we should carefully compare the experimental details to figure out the inherent reasons. Simply distrusting and arbitrarily “anticipating” will not expand our knowledge and are thus not beneficial.

[Editors' note: further revisions were requested prior to acceptance, as described below.]

1) The term "electrostatic gate" seems out of place. The ordinary meaning of a "gate" is to control a flux in an all-or-none way. It is very odd to read a sentence like “…the "closed state" of the electrostatic gate allows the permeation of cations.”. Throughout the whole paper, there are no results showing that TMEM16F is only conducting cations or only conducting anions. It seems inappropriate to use the concept of "gate" to represent the control by a change of electrostatic potential. Controlling the cation versus anion selectivity by altering the pore potential (through charge residue mutations) is well documented in pentameric ligand-gated channels. The authors may want to reference those papers to strengthen the main conclusion in this paper.

We thank the reviewers for the critiques. We now have changed all those statements to "modulation/alteration of electrostatic fields" throughout the text. We also have followed the great suggestion to add the reference to other ion channels which have comparable machineries: "Controlling the selectivity between cations and anions by altering the pore potential, referred to as a charge-screening mechanism, is also documented in other ion channels, such as ligand-gated channels and bacterial porins, through charged residues lining the pore and/or through permeating ions within the pore".

2) Synergy of Ca^2+^ and Depolarization: The authors propose that depolarization indirectly modifies permeation by "driving Ca^2+^ into the membrane electric field" (p16). While the conceptual simplicity of this idea is appealing, the authors observe that depolarization still alters the relative permeability even at high (i.e. 1mM) calcium concentrations where the site of action for calcium should be saturated.

We thank the reviewer for the insight. We actually are not sure whether 1 mM Ca^2+^ "saturates" the relative permeability of Cl^-^. A comparison against the trace recorded in 1 mM Gd^3+^ (Figure 5E, wild type) may indicate that the relative Cl^-^ permeability as in Figure 7E (Q559K) can be even higher, although it is technically hard to test it by further increasing Ca^2+^ concentration without introducing other ion species in the solution (the NaCl concentration should be maintained at ~15 mM). We now avoid using the word "saturated", and only make the point that the current is now more carried by Cl^-^ than by Na^+^.

References

1 Yu, Y., Kuan, A. S. and Chen, T. Y. Calcium-calmodulin does not alter the anion permeability of the mouse TMEM16A calcium-activated chloride channel. J Gen Physiol 144, 115-124, doi:10.1085/jgp.201411179 (2014).

2 Sapar, M. L. et al. Phosphatidylserine Externalization Results from and Causes Neurite Degeneration in *Drosophila*. Cell Rep 24, 2273-2286, doi:10.1016/j.celrep.2018.07.095 (2018).

3 Forschbach, V. et al. Anoctamin 6 is localized in the primary cilium of renal tubular cells and is involved in apoptosis-dependent cyst lumen formation. Cell Death Dis 6, e1899, doi:10.1038/cddis.2015.273 (2015).

4 Henkel, B. et al. Co-expression of anoctamins in cilia of olfactory sensory neurons. Chem Senses 40, 73-87, doi:10.1093/chemse/bju061 (2015).

5 Delling, M., DeCaen, P. G., Doerner, J. F., Febvay, S. and Clapham, D. E. Primary cilia are specialized calcium signalling organelles. Nature 504, 311-314, doi:10.1038/nature12833 (2013).

6 Bricogne, C. et al. TMEM16F activation by Ca(2+) triggers plasma membrane expansion and directs PD-1 trafficking. Sci Rep 9, 619, doi:10.1038/s41598-018-37056-x (2019).

7 Fine, M., Bricogne, C. and Hilgemann, D. Massive surface membrane expansion without involvement of classical exocytic mechanisms. BioRxiv, doi:10.1101/249284 (2018).

8 Jiang, T., Yu, K., Hartzell, H. C. and Tajkhorshid, E. Lipids and ions traverse the membrane by the same physical pathway in the nhTMEM16 scramblase. *eLife* 6, doi:10.7554/*eLife*.28671 (2017).

9 Nguyen, D. M., Chen, L. S., Yu, W. P. and Chen, T. Y. Comparison of ion transport determinants between a TMEM16 chloride channel and phospholipid scramblase. J Gen Physiol, doi:10.1085/jgp.201812270 (2019).

10 Lam, A. K. and Dutzler, R. Calcium-dependent electrostatic control of anion access to the pore of the calcium-activated chloride channel TMEM16A. e*Life* 7, doi:10.7554/*eLife*.39122 (2018).

11 Peters, C. J. et al. The Sixth Transmembrane Segment Is a Major Gating Component of the TMEM16A Calcium-Activated Chloride Channel. Neuron 97, 1063-1077 e1064, doi:10.1016/j.neuron.2018.01.048 (2018).

12 Alvadia, C. et al. Cryo-EM structures and functional characterization of the lipid scramblase TMEM16F. BioRxiv 455261, doi:https://doi.org/10.1101/455261 (2018).

13 Ye, W. et al. Phosphatidylinositol-(4, 5)-bisphosphate regulates calcium gating of small-conductance cation channel TMEM16F. Proc Natl Acad Sci U S A 115, E1667-E1674, doi:10.1073/pnas.1718728115 (2018).

14 Dang, S. et al. Cryo-EM structures of the TMEM16A calcium-activated chloride channel. Nature 552, 426-429, doi:10.1038/nature25024 (2017).

15 Paulino, C., Kalienkova, V., Lam, A. K. M., Neldner, Y. and Dutzler, R. Activation mechanism of the calcium-activated chloride channel TMEM16A revealed by cryo-EM. Nature 552, 421-425, doi:10.1038/nature24652 (2017).

16 Paulino, C. et al. Structural basis for anion conduction in the calcium-activated chloride channel TMEM16A. *eLife* 6, doi:10.7554/*eLife*.26232 (2017).

17 Bushell, S. et al. The structural basis of lipid scrambling and inactivation in the endoplasmic reticulum scramblase TMEM16K. BioRxiv 447417, doi:https://doi.org/10.1101/447417 (2018).

18 Yang, H. et al. TMEM16F forms a Ca^2+^-activated cation channel required for lipid scrambling in platelets during blood coagulation. Cell 151, 111-122, doi:10.1016/j.cell.2012.07.036 (2012).

19 Yu, K. et al. Identification of a lipid scrambling domain in ANO6/TMEM16F. *eLife* 4, e06901, doi:10.7554/*eLife*.06901 (2015).